# CosPGD: AN EFFICIENT WHITE-BOX ADVERSARIAL ATTACK FOR PIXEL-WISE PREDICTION TASKS

## ABSTRACT

While neural networks allow highly accurate predictions in many tasks, their lack of robustness towards even slight input perturbations hampers their deployment in many real-world applications. White-box adversarial attacks such as the seminal *projected gradient descent* (PGD) offer an effective means to evaluate the model robustness and dedicated solutions have been proposed for example for attacks on semantic segmentation or on optical flow. To streamline the evaluation process, we propose an efficient white-box adversarial attack, termed CosPGD, that can be applied to any pixel-wise prediction task in a unified setting. To this end, CosPGD employs a simple loss scaling based on the cosine similarity between the distributions over the predictions and ground truth (or target, for targeted attacks). This leads to efficient evaluations of a model's robustness for pixelwise classification as well as regression models, providing new insights into their performance at earlier attack stages. We outperform the *SotA* on semantic segmentation attacks in our experiments on PASCAL VOC2012 and CityScapes. Further, we showcase CosPGD's versatility by evaluating optical flow as well as image restoration models. We provide code for the CosPGD algorithm and example usage at `https://anonymous.4open.science/r/cospgd-iclr2024-909/`.

## 1 INTRODUCTION

Deep Neural Networks (DNNs) have been gaining popularity for estimating solutions to various complex tasks including numerous vision tasks like classification (Krizhevsky et al., 2012; He et al., 2015; Xie et al., 2016; Liu et al., 2022), semantic segmentation (Ronneberger et al., 2015; Zhao et al., 2017), or disparity (Li et al., 2020) and optical flow (Fischer et al., 2015; Ilg et al., 2016; Teed and Deng, 2020) estimation, due to their overall precise predictions. However, DNNs are inherently black-box function approximators (Buhrmester et al., 2019), known to find shortcuts to map the input to a target (Geirhos et al., 2020) or to learn biases (Geirhos et al., 2018). Thus, we have limited information on the quality of representations learned by the network and their robustness.

An adversarial attack adds a crafted, small (epsilon-sized) perturbation to the input of a neural network that aims to alter the prediction, thus assessing a network's robustness as in the benchmarks (Croce et al., 2021; Jung et al., 2023). Due to the practical relevance to evaluate and analyze DNN models, such attacks have been extensively studied (Goodfellow et al., 2014; Kurakin et al., 2017; Wong et al., 2020b; Madry et al., 2017; Moosavi-Dezfooli et al., 2015; Kurakin et al., 2016). Single step attacks such as FGSM (Goodfellow et al., 2014) are most efficient but often less effective and reliable than multistep versions such as *projected gradient descent* (PGD) (Kurakin et al., 2017) and subsequent attacks (Schrodi et al., 2022; Croce and Hein, 2020; 2021) for image classification.

Existing approaches predominantly focus on attacking image classification models. However, arguably, the robustness of models for pixel-wise prediction tasks is highly relevant for many safety-critical applications such as motion estimation in autonomous driving or image segmentation. The application of existing attacks to pixel-wise prediction tasks such as semantic segmentation or optical flow estimation is possible in principle (e.g. as in Arnab et al. (2017)), albeit carrying only limited information since the pixel-specific loss information is not fully leveraged. In Figure 1, we illustrate this effect for a targeted attack on optical flow estimation and show that classical classification attacks such as PGD (see Figure 1(e)) can only fool the network predictions to some extent: PGD tends to only fit the target (all zeros, i.e. white) in parts of the optical flow, while CosPGD is more coherent.

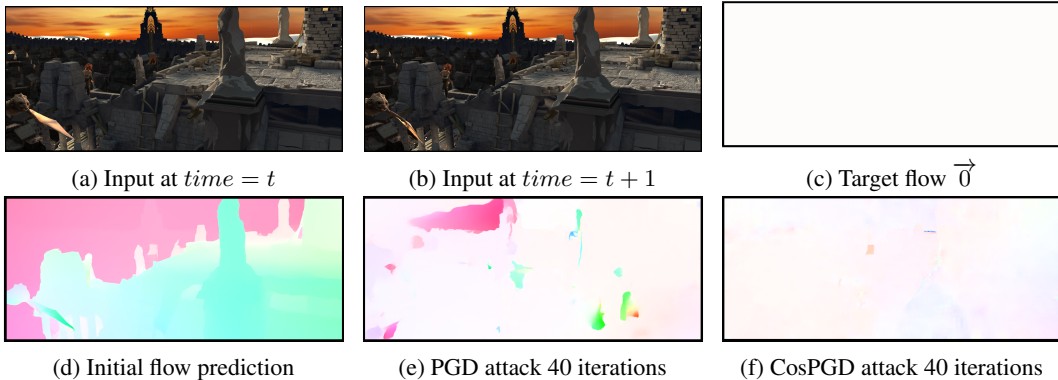

(a) Input at $time = t$     (b) Input at $time = t + 1$     (c) Target flow $\overrightarrow{0}$

(d) Initial flow prediction     (e) PGD attack 40 iterations     (f) CosPGD attack 40 iterations

Figure 1: Optical flow predictions using RAFT (Teed and Deng, 2020) on Sintel (Butler et al., 2012; Wulff et al., 2012) validation. (a) and (b) show two consecutive frames for which the initial optical flow in (d) was predicted. The results of attacking the model with target $\overrightarrow{0}$ (c) are depicted in (e) for PGD and (f) for CosPGD. For the same perturbation magnitude and number of iterations, the proposed CosPGD alters the estimated optical flow more strongly and brings it closer to target (c).

For semantic segmentation, Gu et al. (2022) showed that harnessing pixel-wise information for adversarial attacks leads to much stronger attacks. They argue that, during the attack, the loss to be backpropagated needs to be altered such that already flipped pixel predictions are less important for the gradient computation. Thus, SegPGD (Gu et al., 2022) makes a binary decision for each pixel based on the classification result at this location in order to weigh the attack loss for incorrect and correct model predictions individually. While this is intuitive for semantic segmentation, it can not extend to pixel-wise regression tasks by definition. Furthermore, SegPGD has to fade back in the loss of already incorrectly predicted pixels over time, since otherwise the overall gradient vanishes or becomes instable (Gu et al., 2022). We hypothesize that this leads to SegPGD needing more iterations than necessary. Soft decisions on each pixel's impact to the overall attack update are therefore beneficial to increase attack efficiency.

In this work, we propose CosPGD, an efficient white-box adversarial attack that considers the cosine-similarity between the prediction and target for each pixel. Due to its principled formulation, CosPGD can be used for a wide range of pixel-wise prediction tasks beyond semantic segmentation. Figure 1(f) shows its effect on optical flow estimation, where it can fit the target at almost all locations. Since it leverages the posterior distribution of the prediction for loss computation, it can significantly outperform SegPGD on semantic segmentation. This additionally enables CosPGD to gauge the model's robustness correctly with significantly fewer iterations than SegPGD, thus making the attack efficient. The main contributions of this work are as follows:

- We propose an efficient white-box adversarial attack, CosPGD that can be used to attack all pixel-wise prediction tasks, including semantic segmentation as well as optical flow prediction, and thus allows for an efficient evaluation of their robustness in a unified setting.

- For semantic segmentation, we compare CosPGD to the recently proposed SegPGD which also uses pixel-wise information for generating attacks but can only be applied to semantic segmentation models. CosPGD outperforms SegPGD by a significant margin.

- The proposed CosPGD can be used as a *targeted* attack and as a *non-targeted* attack. We provide implementations for both $l_2$ and $l_\infty$ bounded CosPGD attacks to allow for a wide range of evaluation scenarios.

- To demonstrate CosPGD's versatility, we also evaluate it on optical flow estimation and image restoration in several settings and on several datasets and compare to previous attacks.

## 2  RELATED WORK

The vulnerability of DNNs to adversarial attacks was first explored in Goodfellow et al. (2014) for image classification, proposing the Fast Gradient Sign Method (FGSM). FGSM is a single-step (one

iteration) white-box adversarial attack that perturbs the input in the direction of its gradient, generated from backpropagating the loss, with a small step size, such that the model prediction becomes incorrect. Due to its fast computation, it is still a widely used approach. Numerous subsequent works have been directed towards generating effective adversarial attacks for diverse tasks including NLP (Morris et al., 2020; Ribeiro et al., 2018; Iyyer et al., 2018), or 3D tasks (Zhang et al., 2021; Sun et al., 2021). Yet, the high input dimensionality of image classification models results in the striking effectiveness of adversarial attacks in this field (Goodfellow et al., 2014; Jia et al., 2022). A vast line of work has been dedicated to assessing the quality and robustness of representations learned by the network, including the curation of dedicated evaluation data for particular tasks (Kang et al., 2019; Hendrycks and Dietterich, 2019; Hendrycks et al., 2019) or the crafting of effective adversarial attacks. These adversarial attacks can be image-wide or localized in a small region or patch. These perturbations are in a small region of the image and are called Patch Attacks (e.g. Brown et al. (2017)),while methods such as proposed in Goodfellow et al. (2014); Kurakin et al. (2017); Madry et al. (2017); Wong et al. (2020b); Moosavi-Dezfooli et al. (2015); Croce and Hein (2020); Andriushchenko et al. (2020); Carlini and Wagner (2017); Rony et al. (2019); Dong et al. (2018) argue in a Lipschitz continuity motivated way that a robust network's prediction should not change drastically if the perturbed image is within the epsilon-ball of the original image and thus optimize attacks globally within the epsilon neighborhood of the original input. Our proposed CosPGD approach follows this line of work.

White-box attacks assume full access to the model and its gradients (Goodfellow et al., 2014; Kurakin et al., 2017; Madry et al., 2017; Wong et al., 2020b; Gu et al., 2022; Moosavi-Dezfooli et al., 2015; Rony et al., 2023a; Dong et al., 2018) while black-box attacks optimize the perturbation in a randomized way (Andriushchenko et al., 2020; Ilyas et al., 2018; Qu et al., 2023). The proposed CosPGD derives its optimization objective from Projected Gradient Descent PGD (Kurakin et al., 2017) and is a white-box attack.

Further, one distinguishes between *targeted attacks* (e.g.Wong et al. (2020a); Gajjar et al. (2022); Schmalfuss et al. (2022)) that turn the network predictions towards a specific target and *untargeted attacks* (or non-targeted attacks) that optimize the attack to cause any incorrect prediction. PGD (Kurakin et al., 2017), and CosPGD by extension, allows for both settings (Vo et al., 2022).

While previous attacks predominantly focus on classification tasks, only a few approaches specifically address the analysis of pixel-wise prediction tasks such as semantic segmentation, optical flow, or disparity estimation. For example, PCFA (Schmalfuss et al., 2022) was applied to the estimation of optical flow and specifically minimizes the average end-point error ($AEE$) to a target flow field. A notable exception of pixel-wise white-box adversarial attack is proposed in Gu et al. (2022). The recent SegPGD attack could showcase the importance of pixel-wise attacks for semantic segmentation. In this work, we propose CosPGD to provide a principled and efficient adversarial attack, that can be applied to a wide range of pixel-wise prediction tasks.

Similar to SegPGD, the here proposed CosPGD is based on the optimization formulated in Projected Gradient Descent (PGD) (Kurakin et al., 2017). PGD in its formulation is very similar to FGSM, i.e. it aims to increase the network's loss for an image by adding epsilon-bounded noise. Yet, it is significantly more expensive to optimize than FGSM since it is allowed not one but multiple optimization steps. We explain PGD in more detail in Section 3. SegPGD (Gu et al., 2022) extends upon PGD for semantic segmentation by considering the loss per pixel. For effective optimization, it splits the model's predicted segmentation mask into correctly classified and incorrectly classified pixels, by comparing it to the ground truth segmentation mask. Thus, the loss is scaled over the iterations and the attack does not continue to increase the loss on already flipped pixel labels. While SegPGD improves upon previous adversarial attacks, it is limited to pixel-wise *classification* tasks (i.e. semantic segmentation) by definition and cannot be extended to regression-based tasks like disparity estimation or optical flow estimation. Thus, we propose CosPGD. CosPGD uses the pixel-wise cosine similarity between the distribution over the predictions and the distribution over the targets to scale the loss for each pixel so that it can be applied to classification and regression tasks in a principled way. Further, the cosine similarity can be evaluated on the prediction scores for pixel-wise classification tasks and thereby leverage even more information from the network. Thus, CosPGD outperforms SegPGD by a significant margin when attacking semantic segmentation models while preserving the efficiency of SegPGD and extending it to other pixelwise prediction tasks.

---

**Algorithm 1** Algorithm for generating adversarial examples using CosPGD.

---

**Require:** model $f_{\text{net}}(\cdot)$, clean samples $\boldsymbol{X}^{\text{clean}}$, perturbation range $\epsilon$, step size $\alpha$, attack iterations $T$, ground truth/target $\boldsymbol{Y}$

$\boldsymbol{X}^{\text{adv}_0} = \boldsymbol{X}^{\text{clean}} + \mathcal{U}(-\epsilon, +\epsilon)$        ▷ initialize adversarial example and clip to valid $l_\infty$ or $l_2$ bound

**for** t ← 0 to T-1 **do**        ▷ loop over attack iterations

     $P = f_{\text{net}}(\boldsymbol{X}^{\text{adv}_t})$        ▷ make predictions

     cossim ← $CosineSimilarity(\Psi^*(P), \Psi'(\boldsymbol{Y}))$        ▷ compute cosine similarity

     if targeted attack:

         cossim ← 1 − cossim        ▷ punish dissimilarity to target

         $\alpha \leftarrow -\alpha$        ▷ opposite direction for targeted attack

     $L_{\cos} \leftarrow$ cossim $\cdot L(P, \boldsymbol{Y})$        ▷ scaling the pixel-wise loss for sample updates

     $\boldsymbol{X}^{\text{adv}_{t+1}} \leftarrow \boldsymbol{X}^{\text{adv}_t} + \alpha \cdot sign(\nabla_{\boldsymbol{X}^{\text{adv}_t}} L_{\cos})$        ▷ update adversarial examples

     $\delta \leftarrow \phi^\epsilon(\boldsymbol{X}^{\text{adv}_{t+1}} - \boldsymbol{X}^{\text{clean}})$        ▷ clip $\delta$ to valid $l_\infty$ or $l_2$ bound

     $\boldsymbol{X}^{\text{adv}_{t+1}} = \phi^\epsilon(\boldsymbol{X}^{\text{clean}} + \delta)$        ▷ add $\delta$ to $\boldsymbol{X}^{\text{clean}}$ and clip into valid image range

**end for**

$P = f_{\text{net}}(\boldsymbol{X}^{\text{adv}_T})$        ▷ make predictions on adversarial examples

---

## 3 METHOD

CosPGD is an iterative white-box attack that uses the pixel-wise cosine similarity to generate strong adversarial examples. It effectively extends PGD to all pixel-wise prediction tasks using the same attack step as PGD, given by Equation 1 and Equation 2 and in Kurakin et al. (2017); Gu et al. (2022).

$$\boldsymbol{X}^{\text{adv}_{t+1}} = \boldsymbol{X}^{\text{adv}_t} + \alpha \cdot \text{sign}\nabla_{\boldsymbol{X}^{\text{adv}_t}} L(f_{\text{net}}(\boldsymbol{X}^{\text{adv}_t}), \boldsymbol{Y}), \qquad \delta = \phi^\epsilon(\boldsymbol{X}^{\text{adv}_{t+1}} - \boldsymbol{X}^{\text{clean}}) \tag{1}$$

$$\boldsymbol{X}^{\text{adv}_{t+1}} = \phi^r(\boldsymbol{X}^{\text{clean}} + \delta) \tag{2}$$

Here, $L(\cdot)$ is a one-differentiable function of the model prediction and the target, which defines the loss the model aims to minimize, $\boldsymbol{X}^{\text{adv}_{t+1}}$ is a new adversarial example for time step $t + 1$, generated using $\boldsymbol{X}^{\text{adv}_t}$, the adversarial example at time step $t$ and initial clean sample $\boldsymbol{X}^{\text{clean}}$. $\boldsymbol{Y}$ is the ground truth label for non-targeted attacks and the target for targeted attacks, $\alpha$ is the step size for the perturbation ($\alpha$ is multiplied by $-1$ for targeted attacks to take a step in the direction of the target), and the function $\phi^\epsilon$ is clipping the $\delta$ in $\epsilon$-ball for $l_\infty$-norm bounded attacks or the $\epsilon$-projection in $l_2$-norm bounded attacks, complying with the $l_\infty$-norm or $l_2$-norm constraints, respectively. $\phi^r$ is clipping the generated example in the valid input range (usually between [0, 1]). $\nabla_{\boldsymbol{X}^{\text{adv}_t}} L(\cdot)$ denotes the gradient of $\boldsymbol{X}^{\text{adv}_t}$ generated by backpropagating the loss and is used to determine the direction of the perturbation step. SegPGD extends this formulation to tensor-valued predictions and labels $\boldsymbol{Y} \in \mathbb{R}^{H \times W \times M}$ for images of size $H \times W$ and categorical $M$ output classes. This allows specifically optimizing perturbations on pixels on which the segmentation is not yet flipped instead of considering all positions equally important.

**CosPGD** The aim of the proposed CosPGD approach is to facilitate the effective application of PGD-like adversarial attacks to pixel-wise prediction tasks in a principled way. To facilitate this, we propose a unified way to scale the loss in classification and regression settings, allowing the attack to appropriately focus on altering predictions that still need to be altered most. Specifically, for non-targeted settings, we aim to penalize pixel-wise in proportion to the pixel-wise predictions' similarity to the ground truth, while also accounting for the decrease in similarity over iterations. For targeted settings, we aim to penalize the dissimilarity of the prediction to the target prediction. We propose to use the cosine similarity as this measure, as it satisfies the desired properties. Cosine similarity provides a measure of similarity between the direction of two vectors and should therefore be well-suited to represent label similarities at the posterior level. Additionally, cosine similarity scales in a fixed range [-1 , 1] so that there are no normalization issues that might affect the stability during optimization.

The cosine similarity between the model predictions and target (ground truth) is calculated as shown in Equation 3 for each output pixel location:

$$\cos(\overrightarrow{\text{pred}}, \overrightarrow{\text{target}}) = \frac{\overrightarrow{\text{pred}} \cdot \overrightarrow{\text{target}}}{||\overrightarrow{\text{pred}}|| \cdot ||\overrightarrow{\text{target}}||}, \tag{3}$$

where $\overrightarrow{\text{pred}}$ is the probability distribution of the predictions of a network $f_{\text{net}}(\cdot)$ for a position, and $\overrightarrow{\text{target}}$ is the distribution over the target predictions or ground truth at the same position. Untargeted CosPGD intends to drive the model's predictions away from the model's intended target (or ground truth). In the case of semantic segmentation, we obtain the distribution $\text{target}$ by generating a *one-hot encoded vector* of the $\text{target}$ and we obtain the distribution over the predictions, by calculating the softmax of the predictions before taking the argmax

$$\overrightarrow{\text{pred}} = softmax(f_{\text{net}}(\boldsymbol{X})), \qquad softmax(x_i) = \frac{\exp(x_i)}{\sum_j \exp(x_j)}. \tag{4}$$

Thus, in Algorithm 1, $\Psi^*$ is always the softmax function, and $\Psi'$ is one-hot encoding in case of semantic segmentation while $\Psi'$ is the softmax function when $\overrightarrow{\text{target}}$ is already a vector. $\boldsymbol{X}^{\text{adv}}$ is initialized to the clean input sample $\boldsymbol{X}^{\text{clean}}$ with added randomized noise in the range $[-\epsilon, +\epsilon]$, $\epsilon$ being the maximum allowed perturbation. Over attack iterations $\boldsymbol{X} = \boldsymbol{X}^{\text{adv}_t}$, the adversarial example generated at iteration $t$, such that $t \in [0, T)$, where $T$ is the total number of attack iterations.

As discussed, we finally propose to scale the pixel-wise loss using cosine similarity (cossim) $\cos(\overrightarrow{\text{pred}}, \overrightarrow{\text{target}})$, such that for the non-targeted setting, pixels where the network predictions are closer to the intended target (ground truth), have a higher similarity (approaching 1) and thus higher loss. Pixels with lower similarity, have a lower loss but are not rendered benign. While for the targeted setting, we consider cosine dissimilarities (refer to Equation 6), and thus, pixels where the network predictions are closer to the target, have higher similarity and thus lower loss, and pixels with lower similarity have a higher loss. Thus, the final loss over all pixels is calculated as shown in Equation 5 & Equation 6. Then this loss is back propagated to obtain gradients over the sample to perform the adversarial attack as shown in Equation 2 using $L_{cos}$ instead of $L$. For non-targeted attacks, we have

$$L_{\cos} = \frac{1}{H \times W} \sum_{H \times W} \cos\left(\overrightarrow{\text{pred}}, \overrightarrow{\text{target}}\right) \cdot L\left(f_{\text{net}}(\boldsymbol{X}^{\text{adv}_t}), \boldsymbol{Y}\right), \tag{5}$$

and for targeted attacks, we have

$$L_{\cos} = \frac{1}{H \times W} \sum_{H \times W} \left(1 - \cos\left(\overrightarrow{\text{pred}}, \overrightarrow{\text{target}}\right)\right) \cdot L\left(f_{\text{net}}(\boldsymbol{X}^{\text{adv}_t}), \boldsymbol{Y}\right), \tag{6}$$

where $H$ and $W$ are the height and width of a sample $\boldsymbol{X}$. CosPGD is summarized in Algorithm 1.

**Loss scaling in previous approaches** When optimizing $\delta$ for an adversarial attack, Gu et al. (2022) argue that pixels which are already misclassified by the model are less relevant than pixels correctly classified by the model, because the intention of the attack is to make the model misclassify as many pixels as possible while perturbing the $\delta$ inside the $\epsilon$-ball. As a consequence, they make a hard decision based on each pixels argmax prediction as of whether it is taken into account for attack computation. In consequence, as the number of misclassified pixels increases, the attack loses effectiveness if it only focuses on correctly classified pixels. As remedy, Gu et al. (2022) propose to scale the loss over iterations such that the scaling of the loss for correctly classified pixels is inversely proportional to the scaling of the loss for the incorrectly classified pixels. This avoids the concern of the attack becoming benign after a few iterations, yet it fades out the effect of SegPGD and may reduce its efficiency. CosPGD, operating on continuous predictions, does not require such heuristic.

Furthermore, splitting the pixels into two categories, *correctly* and *incorrectly* classified pixels, limits the applicability of SegPGD to pixel-wise classification tasks (like semantic segmentation) by definition. For pixel-wise regression tasks (like optical flow, or image reconstruction) **there is no absolute measure of correctness**, so SegPGD can not be applied.

Lastly, comparing the pixel-wise labels after taking the argmax only provides limited information from the network prediction. CosPGD expands the scope of the categories such that it can encompass the similarity between the predictions and the label to leverage it for more effective update steps, resulting in improved efficiency after comparable numbers of iterations.

## 4 EXPERIMENTS

To demonstrate the wide applicability of CosPGD, we conduct our experiments on distinct downstream tasks: semantic segmentation, optical flow estimation, and image restoration. For semantic

segmentation, we compare CosPGD to SegPGD and PGD, while for optical flow estimation and other tasks (such as image deblurring and image denoising), we compare CosPGD to PGD. We observe that CosPGD is a significantly stronger attack compared to SegPGD and PGD.

When evaluating $l_\infty$-norm constrained attacks, we use the same $\epsilon \approx \frac{8}{255}$ for CosPGD, SegPGD, and PGD. For $\alpha$, we follow Gu et al. (2022) and set the step size to $\alpha = 0.01$ (please refer to Appendix B.2 for an ablation study). Further, when evaluating the $l_2$-norm constraint, we follow (Croce et al., 2020; Wang et al., 2023) and use the same $\epsilon$ for CosPGD, SegPGD, and PGD i.e. $\epsilon \approx \{\frac{64}{255}, \frac{128}{255}\}$ and $\alpha = \{0.01, 0.02\}$. We show in Appendix B.2.1 that CosPGD outperforms both PGD and SegPGD (for segmentation) in the $l_2$-norm constraint settings under all commonly used $\epsilon$ and $\alpha$ values.

**Semantic Segmentation**    We use PASCAL VOC 2012 (Everingham et al., 2012), which contains 20 object classes and one background class, with 1464 training images, and 1449 validation images. We follow common practice (Hariharan et al., 2015; Gu et al., 2022; Zhao, 2019; Zhao et al., 2017), and use work by Hariharan et al. (2011), augmenting the training set to 10,582 images. We evaluate on the validation set. Architectures used for our evaluations are PSPNet (Zhao et al., 2017) and DeepLabV3 (Chen et al., 2017), both with ResNet50 (He et al., 2015) encoders, and UNet (Ronneberger et al., 2015) with a ConvNeXt tiny encoder (Liu et al., 2022). Results are reported in Appendix B.1. We report mean Intersection over Union (mIoU) and mean pixel accuracy (mAcc).

**Optical Flow**    We use RAFT (Teed and Deng, 2020) and follow the evaluation procedure used therein. Evaluations are performed on KITTI2015 (Menze and Geiger, 2015) and MPI Sintel (Butler et al., 2012; Wulff et al., 2012) validation sets. We use the networks pre-trained on FlyingChairs (Dosovitskiy et al., 2015) and FlyingThings (Mayer et al., 2016) and fine-tuned on training datasets of the specific evaluation, as provided by Teed and Deng (2020). For Sintel we report the end-point error ($epe$) on both clean and final subsets, while for KITTI15 we report the $epe$ and $epe$-$f1$-$all$. In Appendix C.3 we compare CosPGD to PCFA across different networks.

**Image Restoration**    For the image de-blurring task we use the GoPro dataset (Nah et al., 2017) as in Chen et al. (2022). The images are split into 2103 training images and 1111 test images. We consider the "Baseline network" and NAFNet as proposed by Chen et al. (2022). For the image restoration tasks we report the $PSNR$ and $SSIM$ scores of the reconstructed images w.r.t. to the ground truth images, averaged over all images. We provide further details in Appendix D.1.

Following Gu et al. (2022); Rony et al. (2023b); Kang et al. (2020), we compute $l_\infty$-norm and $l_2$-norm constrained **non-targeted** attacks for semantic segmentation. For optical flow estimation, we compute $l_\infty$-norm **targeted** attacks and report results using $l_2$-norm in Appendix C.3 to allow for comparison to Schmalfuss et al. (2022). For image restoration, we use $l_\infty$-norm **non-targeted** attacks.

## 4.1    SEMANTIC SEGMENTATION

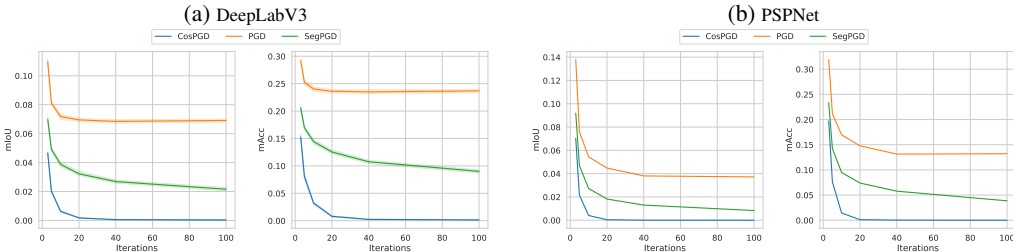

Figure 2: CosPGD versus PGD and SegPGD ($l_\infty$-norm constrained) for semantic segmentation on PASCAL VOC2012 validation set on DeepLabV3 and PSPNet. CosPGD outperforms competing attacks even in early iterations by a large margin. See also Table 4 in Appendix B.

We report the comparison of CosPGD to the recently proposed SegPGD and to PGD in Figure 2. CosPGD yields a much stronger attack compared to PGD or SegPGD. This is consistent

across the number of attack iterations, as CosPGD fools networks at much fewer iterations than SegPGD as measured by both metrics mIoU and mAcc. For example in Figure 3 after 40 attack iterations, all attacks are considerably fooling the network into making incorrect predictions.

However, once the dominant class label is changed by SegPGD or PGD, they do not further optimize over small regions of correct prediction. In contrast, CosPGD successfully fools the model into making incorrect predictions even in these small regions by either swapping the region prediction with an already existing class or forcing the model into predicting a class not existing in that sample.

Greedy approaches like PGD, originally designed for image classification, are able to bring down the $mIoU$ of DeepLabV3 to 6.79%. SegPGD, by naïvely utilizing the pixel-wise segmentation error, deteriorates the model performance further to 2.69%. However, CosPGD is able to fool the network into making incorrect predictions for almost all pixels of the samples, bringing down the model performance to 0.08% after 40 iterations and to approximately 0% after 100 iterations.

Additionally, in Table 4 (Appendix) we observe that at low attack iterations (iterations=3)

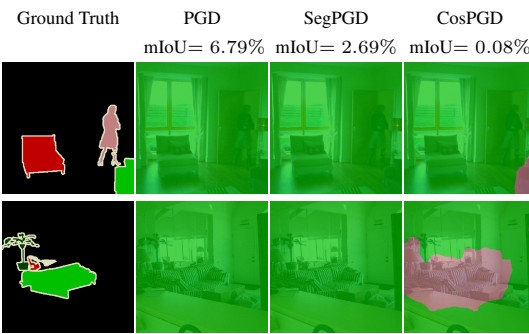

Figure 3: Example predictions of DeepLabV3 on PASCAL VOC 2012 val set after $l_\infty$ PGD, SegPGD, and CosPGD attacks with 40 iters. The ground truth segmentations are given on the left. Both PGD and SegPGD are able to successfully change most of the predicted labels to one of the ground truth labels (here in green). Yet, the region with this label is predicted correctly. Here, the benefit of CosPGD is aptly highlighted as it also changes the prediction in this region to a third class.

SegPGD implies that PSPNet is more adversarially robust than DeepLabV3. However, after more attack iterations (iterations≥5), SegPGD reveals that DeepLabV3 is more robust than PSPNet. Contrary to this, CosPGD even at low attack iterations correctly predicts DeepLabV3 to be more robust than PSPNet. This is an insight that CosPGD provides with considerably **fewer iterations, thus lower overall computation time**. Compute costs per iteration are comparable, see Table 2 (Appendix).

Furthermore, the improved performance of CosPGD is not limited to $l_\infty$-norm constrained attacks that were observed in Figures 2 & 3. As shown in Figure 4, it also extends to $l_2$-norm constrained attacks, where CosPGD again outperforms both SegPGD and PGD across attack iterations. Moreover, the gap in performance of the three adversarial attacks significantly increases when increasing the number of attack iterations. This demonstrates that CosPGD is able to utilize the increase in attack iterations best and highlights the significance of scaling the pixel-wise loss with the cosine similarities rather than using a heuristical scaling as in SegPGD.

Figure 4: $l_2$-norm constrained CosPGD vs. PGD and SegPGD for semantic segmentation over PASCAL VOC2012 val set for DeepLabV3. See Appendix B.2 for results for more $\alpha$ and $\epsilon$ values.

Thus, we successfully demonstrate the benefit of the proposed attack over existing adversarial attacks for semantic segmentation. We provide more results on $l_\infty$-norm and $l_2$-norm constrained non-targeted adversarial attacks for semantic segmentation using UNet (Ronneberger et al., 2015) with ConvNeXt backbone on **CityScapes** (Cordts et al., 2016) in Appendix B.1, further confirming the benefit of CosPGD.

Additionally, we ablate over the attack step size $\alpha$ for $l_\infty$-norm constrained attacks on DeepLabV3 using PASCAL VOC2012 validation dataset in Appendix B.2.2 and over multiple attack step size $\alpha$ and permissible perturbation $\epsilon$ for $l_2$-norm constrained attacks on DeepLabV3 using PASCAL VOC2012 validation dataset in Appendix B.2. Thus demonstrating the versatility of CosPGD.

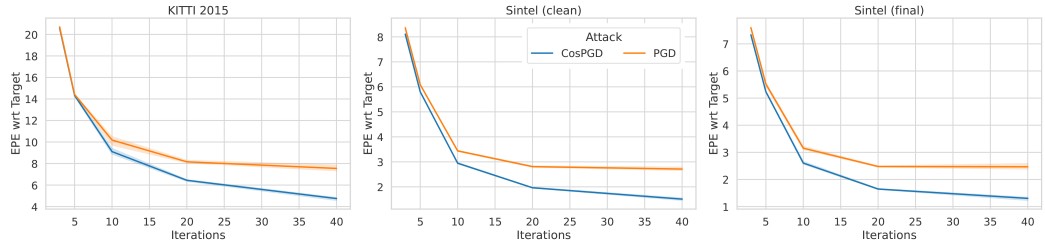

Figure 5: Comparison of performance of CosPGD to PGD for optical flow estimation over KITTI-2015 (left) and Sintel (clean → centre and final → right) validation datasets as $l_\infty$-norm constrained targeted attacks using RAFT. We observe that CosPGD as an attack is stronger than PGD. We also report these results in Table 6 in Appendix C.1.

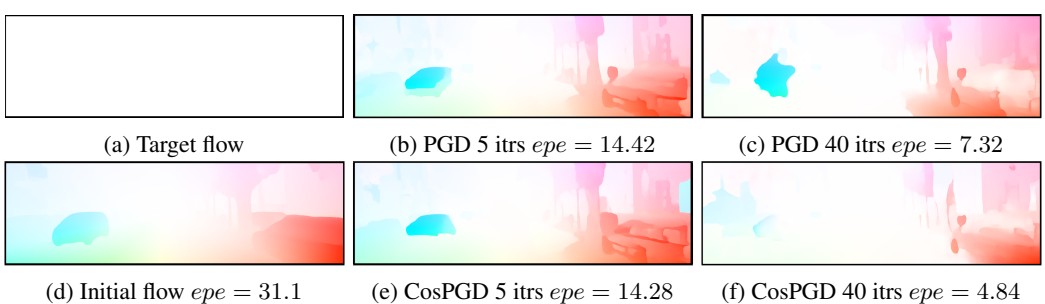

Figure 6: Comparing PGD and CosPGD as a targeted $l_\infty$-norm constrained attack on RAFT using KITTI15 validation set over various iterations. (a) shows the targeted prediction, a $\vec{0}$, and (d) shows the initial optical flow estimation by the network before adversarial attacks. EPEs between the target and the final prediction are reported, thus lower epe is better. (b) and (c) show flow predictions after PGD attack over 5 and 40 iterations respectively, while figures (e) and (f) show flow predictions after CosPGD attack over 5 and 40 iterations respectively. CosPGD significantly changes the overall structure of the optical flow field, bringing is visibly closer to target (a).

## 4.2    OPTICAL FLOW

For optical flow, we explore the $l_\infty$-norm constrained targeted setting. A comparison of CosPGD to PGD is shown in Figure 5. Here we quantitatively observe the better performance of CosPGD compared to PGD. As this is the targeted setting, we intend to close the gap between the target prediction and the model predictions, thus a lower $epe$ of the model prediction w.r.t the target prediction is desired. As the attack iterations increase, across datasets, CosPGD is able to significantly fool the network into making predictions closer to the target, bringing down the $epe$ to as low as 1.55 for Sintel (final). We qualitatively observe in Figure 6 that the initial optical flow estimation by the model (which is very different from the target) is only moderately changed when the model is attacked with a strong adversarial attack like PGD. As the attack was originally designed for classification tasks, the model is not significantly fooled even as the intensity of the attack is increased to 40 iterations. Figure 6(b), shows qualitatively that the model predictions are not significantly different from the initial predictions. The shape of the moving car is preserved to a considerable extent. The limited effectiveness of the PGD attack is further highlighted on increasing the attack strength to 40 iterations (see Figure 6(c)). Here, some initial predictions are still preserved, for example the bark of the tree. This is in contrast to when the model is attacked using CosPGD, a method that utilizes pixel-wise information. In Figure 6(e), we observe that even at a low intensity of the attack (5 iterations), the model predictions are significantly different from the initial predictions, especially in the background and the shape of the moving car. The model is incorrectly predicting the motion of the pixels around the moving car. At high attack intensity, as shown in Figure 6(f) with 40 iterations, the model's optical flow predictions are significantly inaccurate and exceedingly different from the initial predictions and very close to the target of $\vec{0}$. The model fails to differentiate the moving car from its background, moreover, the bark of the tree has completely vanished. In a

real-world scenario, this vulnerability of the model to a relatively small perturbation ($\epsilon = \frac{8}{255}$) could be hazardous. CosPGD provides us with this new insight. A similar observation is made for the Sintel dataset as shown in Figure 1. The benefit of CosPGD over PGD for optical flow can be quantitatively seen in Figure 5 and Table 6 provided in the supplementary material. In addition, we provide results comparing CosPGD to PGD as a $l_\infty$-constrained non-targeted attack for optical flow estimation in Appendix C.2. We also provide a comparison to PCFA (Schmalfuss et al., 2022) in Appendix. C.3.

### 4.3 IMAGE RESTORATION

CosPGD can be very useful in predicting a new model's robustness efficiently. To demonstrate its versatility, we consider the new vision transformer-based image restoration model NAFNet (Chen et al., 2022). NAFNet outperforms Restormer (Zamir et al., 2022) for image restoration tasks like image de-blurring and image denoising on clean data, thus implying that NAFNet learns good representations.

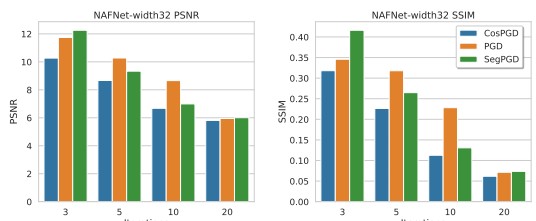

Here, we compare the adversarial robustness of these networks and discuss interesting findings. Figure 7 depicts results for NAFNet on image deblurring of the GoPro dataset images. We observe that CosPGD is a significantly stronger attack than both PGD and SegPGD. To enable the applicability of SegPGD on this task, we implement SegPGD by comparing the equality of the pixel values to use their proposed loss for comparison.

Figure 7: Non-targeted $l_\infty$-norm constrained CosPGD, PGD, and SegPGD attacks on NAFNet, recently proposed by Chen et al. (2022) as the state-of-the-art network for image de-blurring on the Go-Pro dataset.CosPGD significantly outperforms the other attacks. Lower PSNR and SSIM indicate a worse restoration and thus a stronger attack.

We observe that at low number of attack iterations (3 attack iterations) it performs significantly worse than PGD, thus demonstrating its limitation on this task. Interestingly, after 5 attack iterations, SegPGD is stronger than PGD. This exhibits the strength of pixel-wise scaling of the loss. Nonetheless, CosPGD is able to provide us with both wide applicability and an even stronger attack while being more efficient.

We provide further discussion and results on Restormer (Zamir et al., 2022) and the "Baseline network" (Chen et al., 2022) in Appendix D.1. We additionally discuss results on the image denoising task in Appendix D.2.

## 5 CONCLUSION

In this work, we demonstrated across different downstream tasks and architectures that our proposed adversarial attack, CosPGD, is significantly more effective than other existing and commonly used adversarial attacks on several pixel-wise prediction tasks. We provide a new algorithm for evaluating the adversarial robustness of models on pixel-wise tasks. By comparing CosPGD to attacks like PGD, which were originally proposed for image classification tasks, we expanded on the work by Gu et al. (2022) and highlighted the need and effectiveness of attacks specifically designed for pixel-wise prediction tasks beyond segmentation. We illustrated the intuition behind using cosine similarity as a measure for generating stronger adversaries and leveraging more information from the model and backed it with experimental results from different downstream tasks. This further highlights the simplicity and principled formulation of CosPGD, thus making it applicable to a wide range of pixel-wise prediction tasks and in principle extendable to all Lipschitz continuous bounds as a targeted or a non-targeted attack.

**Limitations**    There are settings, especially for non-targeted attacks, where approaches like pixel-wise PGD would work at par with CosPGD as the *epe* can be increased equally well by either changing all pixel-wise regression estimates slightly (sophisticated attack like CosPGD) or by changing only a few of them drastically (brute force attacks like PGD). We discuss this further in detail in Appendix E.

## ETHICS STATEMENT

We have carefully read the ICLR 2024 Code of Ethics and confirm that we adhere to it. The proposed work is original and novel. To the best of our knowledge, all literature used in this work has been referenced correctly. Our work did not involve any human subjects and does not pose a threat to humans or the environment. Adversarial attacks are time and computation exhaustive. And thus, our proposed adversarial attack, CosPGD helps in this regards as it can provide new insights into a model's robustness and vulnerabilities with much less time and thus computation.

## REPRODUCIBILITY

Our work is completely reproducible. We use publicly available code bases for our work. The code for CosPGD attack is available at `https://anonymous.4open.science/r/cospgd-iclr2024-909/` with instructions to reproduce the results. The provided code can be used in a plug-and-play fashion with any pixel-wise prediction task. The code will be made publicly available upon acceptance. We provide further details in Appendix A.1.1

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

# CosPGD: an efficient and unified white-box adversarial attack for pixel-wise prediction tasks

## Supplementary Material

We include the following information in the supplementary material:

- Section A Additional Details:
    - Section A.1 : Hardware details
    - Section A.1.1: Implementation details including code and example usage.
    - Section A.1.3: We provide additional experimental details for the image deblurring experiments.
    - Section A.1.4: We compare the time taken by different adversarial attacks for different tasks.
    - Section A.1.2: Details on calculating *epe-f1-all*.
- Section B: Semantic Segmentation Additional Results:
    - Section B.1: We provide extra $l_\infty$-norm and $l_2$-norm constrained non-targeted adversarial attack results from Semantic Segmentation using the UNet architecture with ConvNeXt backbone on the **CityScapes** dataset (Cordts et al., 2016).
    - Section B.2: We provide an ablation study on attack step size $\alpha$ and $\epsilon$ for $l_2$-norm bounded for non-targeted adversarial attack results from Semantic Segmentation using DeepLabV3 on the PASCAL VOC 2012 dataset.
    - Section B.2.2 We provide an ablation study on attack step size $\alpha$ for $l_\infty$-norm bounded for non-targeted adversarial attack results from Semantic Segmentation using DeepLabV3 on the PASCAL VOC 2012 dataset.
    - Section B.3: We report results from Figure 2 in a tabular form.
    - Section B.4: We report the results of adversarial training for semantic segmentation.
- Section C Optical Flow Additional Results:
    - Section C.1: We report results from Figure 5 in a tabular form.
    - Section C.2: We provide extra results comparing CosPGD to PGD as a $l_\infty$-norm constrained non-targeted adversarial attack for optical flow estimation.
    - Section C.3: We provide a comparison to the $l_2$-constrained PCFA (Schmalfuss et al., 2022), which is dedicated for optical flow.
- Section D: Image Restoration Results:
    - Section D.1: We report the findings on the adversarial robustness of many recently proposed transformer-based image deblurring models.
    - Section D.2: We report the results on many recently proposed transformer-based image denoising models.

In Table 1, we provide a look-up table for all experiments considered in this supplementary material. We provide details on the downstream tasks, models, targeted and non-targeted attack settings, and $l_\infty$-norm constrained and $l_2$-norm constrained settings considered respectively do demonstrate the wide-applicability of CosPGD.

Table 1: All considered experiments in this work.

| Downstream Task | Networks | Dataset | Non-targeted Attack | | Targeted Attack | |
|---|---|---|---|---|---|---|
| | | | $l_\infty$-norm constraint | $l_2$-norm constraint | $l_\infty$-norm constraint | $l_2$-norm constraint |
| Semantic Segmentation | DeepLabV3 PSPNet UNet | PASCAL VOC 2012, Cityscapes | Sec. B.2.2 | Sec. B.2.1 | | |
| Optical Flow Estimation | RAFT PWCNet, GMA, SpyNet | KITTI 2015, Sintel (clean and final) | Sec. C.2 | | Sec. C | Sec. C.3 |
| Image Deblurring | Restormer, Baseline net, NAFNet | GoPro | Sec. D.1 | | | |
| Image Denoising | Baseline net, NAFNet | SSID | Sec. D.2 | | | |

## A  APPENDIX

### A.1  FURTHER EXPERIMENTAL DETAILS ON HARDWARE AND METRICS

**Semantic Segmentation.**  For the experiments on DeepLabV3, we used NVIDIA Quadro RTX 8000 GPUs. For PSPNet, we used NVIDIA A100 GPUs. For the experiments with UNet, we used NVIDIA GeForce RTX 3090 GPUs.

**Image Restoration.**  For the experiments on Image de-blurring tasks, we used NVIDIA GeForce RTX 3090 GPUs.

A single GPU was used for each run.

**Optical Flow Estimation.**  We used NVIDIA V100 GPUs, a single GPU was used for each run.

#### A.1.1  CODE FOR THE ATTACK

The code for the functions used for generating adversarial samples using CosPGD and other considered adversarial attacks in the main paper is available at `https://anonymous.4open.science/r/cospgd-iclr2024-909/`.

Additionally, we provide sample code demonstrating the usage of the packages for a UNet-like architecture with detailed instructions at `https://anonymous.4open.science/r/cospgd-iclr2024-909/`.

#### A.1.2  CALCULATING EPE-F1-ALL

Following the work by Teed and Deng (2020), $f1 - all$ is calculated by averaging $out$ over all the predicted optical flows. $out$ is calculated using Equation equation 7,

$$out = epe > 3.0 \cup \frac{epe}{mag} > 0.05 \tag{7}$$

Where, $mag = \sqrt{flow\ ground\ truth^2}$ and $epe$ is the Euclidean distance between the two vectors.

#### A.1.3  IMAGE DEBLURRING EXPERIMENTAL DETAILS

Chen et al. (2022) simplify a transformer-based architecture Restormer (Zamir et al., 2022) for image restoration tasks and first propose a simplified architecture as a Baseline network, and then improve upon it with intuitions backed by reasoning and ablation studies to propose Non-linear Activation Free Networks abbreviated as NAFNet. In this work, we perform adversarial attacks on both the Baseline network and NAFNet.

**Dataset.**  Similar to Chen et al. (2022), for the image de-blurring task, we use the GoPro dataset (Nah et al., 2017) which consists of 3124 realistically blurry images of resolution $1280 \times 720$ and corresponding ground truth sharp images obtained using a high-speed camera. The images are split into 2103 training images and 1111 test images. For the image denoising task, we use the Smartphone Image Denoising Dataset (SSID) (Abdelhamed et al., 2018). This dataset consists of 160 noisy images taken from 5 different smartphones and their corresponding high-quality ground truth images.

**Metrics.**  For both the image restoration tasks, we report the $PSNR$ and $SSIM$ scores of the reconstructed images w.r.t. to the ground truth images, averaged over all images. $PSNR$ stands for Peak Signal-to-Noise ratio, a higher $PSNR$ indicates a better quality image or an image closer to the image to which it is being compared. $SSIM$ stands for Structural similarity (Wang et al., 2004).

#### A.1.4  COMPARING TIME TAKEN BY DIFFERENT ADVERSARIAL ATTACKS

Following, we report the approximate time taken by each attack in minutes. Please note, this time includes time taken for data-loading and saving of experimental results including images. For a given task, network, and dataset, the time taken by different attacks is comparable and representative of the time taken by the attacks as they followed the same attack procedures. We observe in Table 2 that the

difference in time taken by the different attacks at the same number of iterations is negligible. This is because operations like one-hot encoding and softmax take negligible time.

Thus, the ability of CosPGD to provide valuable insights into model robustness with significantly less iterations than other methods, as discussed in Section 4.1 and Section 4.3 is a compelling advantage.

Table 2: Comparison of time taken in minutes by different attacks on different downstream tasks for different amount of iterations. The computation times are comparable.

| Task | Network | Dataset | Attack method | Attack iterations | | | | |
| | | | | 3 | 5 | 10 | 2 | 40 |
| | | | | Time (mins) | Time (mins) | Time (mins) | Time (mins) | Time (mins) |
| --- | --- | --- | --- | --- | --- | --- | --- | --- |
| **Semantic Segmenation** | **UNet** | **PASCAL VOC 2012** | **SegPGD** | 28.73 | 36.33 | 58.72 | 88.93 | 163.15 |
| | | | **CosPGD** | 26.67 | 36.75 | 54.45 | 97.08 | 165.35 |
| **Optical Flow** | **RAFT** | **KITTI2012** | **PGD** | 5.90 | 7.73 | 12.23 | 20.98 | 37.45 |
| | | | **CosPGD** | 6.00 | 7.85 | 12.15 | 21.03 | 38.28 |
| | | Sintel (clean + final) | **PGD** | 69.87 | 97.47 | 158.28 | 297.40 | 557.97 |
| | | | **CosPGD** | 73.68 | 102.77 | 160.40 | 287.82 | 602.08 |

# B  SEMANTIC SEGMENTATION

## B.1  SEMANTIC SEGMENTATION WITH UNET ON CITYSCAPES

In the following, we provide extra results on semantic segmentation with UNet on the Cityscapes dataset.

### B.1.1  EXPERIMENTAL SETUP

In this evaluation, we use a UNet architecture (Ronneberger et al., 2015) with a ConvNeXt_tiny encoder(Liu et al., 2022). We extend the implementation from username: mberkay0 (2023)(`www.github.com`) to implement CosPGD, PGD, and SegPGD non-targeted $l_\infty$-norm and $l_2$-norm attacks.

We do these evaluations on the Cityscapes dataset (Cordts et al., 2016). Cityscapes contains a total of 5000 high-quality images and pixel-wise annotations for urban scene understanding. The dataset is split into 2975, 500, and 1525 images for training, validation, and testing respectively. The model is trained on the test split and attacks are evaluated on the validation split.

### B.1.2  EXPERIMENTAL RESULTS AND DISCUSSION

In Figure 8, we report results from the comparison of non-targeted CosPGD to PGD and SegPGD attacks across iterations and across $l_p$-norm constraints: $l_\infty$-norm and $l_2$-norm using UNet architecture with a ConvNeXt tiny encoder on Cityscapes validation dataset. For the $l_\infty$-norm constraint, we use the same $\alpha = 0.01$ and $\epsilon \approx \frac{8}{255}$ as in all previous evaluations. For the $l_2$-norm constraint we follow common work (Croce et al., 2020; Wang et al., 2023) and use the same $\epsilon$ for CosPGD, SegPGD, and PGD i.e. $\epsilon \approx \{\frac{64}{255}, \frac{128}{255}\}$ and $\alpha = \{0.1, 0.2\}$.

Note, SegPGD has been proposed as an $l_\infty$-norm constrained attack. We extend it to the $l_2$-norm constraint merely for complete comparison and curiosity.

We observe in Figure 8 that CosPGD is a significantly stronger attack than both PGD and SegPGD, across iterations and $l_p$-norm constraints, and $\alpha$ and $\epsilon$ values. Even at low attack iterations, it outperforms previous methods significantly, making it particularly efficient. Especially as an $l_2$-norm constrained attack, as shown before in Figure 4 for DeepLabV3 on PASCAL VOC 2012 dataset and discussed before in Section 4.1, as attack iterationsincrease, CosPGD is able to increase the gap in performance quite significantly.

## B.2  ABLATION ON ATTACK STEP SIZE $\alpha$

Further, we provide additional experimental results and ablation studies using DeepLabV3 for semantic segmentation on the PASCAL VOC 2012 validation dataset.

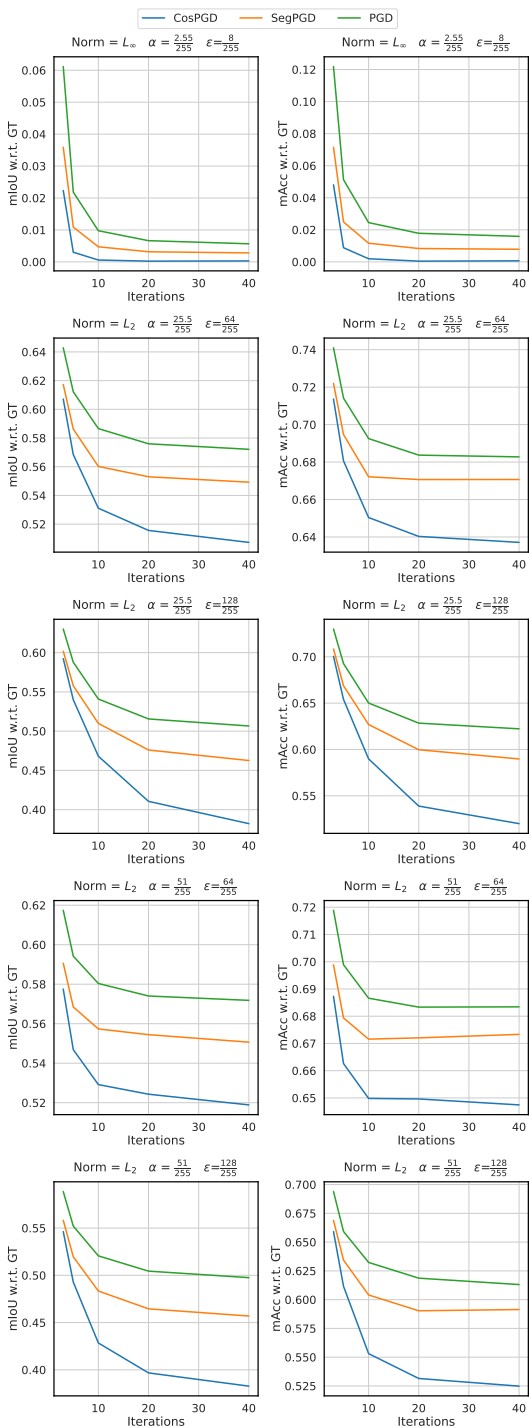

Figure 8: Comparing non-targeted CosPGD to PGD and SegPGD attacks across iterations and $l_p$-norm constraints, and $\alpha$ and $\epsilon$ values using UNet architecture with a ConvNeXt tiny encoder on Cityscapes validation dataset. CosPGD significantly outperforms previous methods by a large margin, even at few attack iterations.

### B.2.1 $l_2$-NORM CONSTRAINED ADVERSARIAL ATTACKS

Further in Figure 9, we extend the results from Figure 4, to report $l_2$-norm constrained attack evaluations on commonly used (Croce et al., 2020; Wang et al., 2023) values of $\epsilon \approx \{\frac{64}{255}, \frac{128}{255}\}$ and $\alpha = \{0.1, 0.2\}$.

Additionally, in Table 3 we provide comparison to C&W Carlini and Wagner (2017) and other $l_2$-norm constrained adversarial attacks with $\alpha$=0.2 and $epsilon \approx \frac{128}{255}$ on PASCAL VOC 2012 validation dataset using DeepLabV3 with a ResNet50 backbone.

Table 3: Comparison of performance of CosPGD to SegPGD, PGD and C&W as a $l_2$-norm constrained attack with $\alpha$=0.2 and $\epsilon \approx \frac{128}{255}$ where applicable for semantic segmentation over PASCAL VOC2012 validation dataset. We observe that CosPGD is a significantly stronger attack compared to all the other attacks for both metrics.

| Network | Attack method | Attack iterations | | | | | | | | | | | |
|---|---|---|---|---|---|---|---|---|---|---|---|---|---|
| | | 3 | | 5 | | 10 | | 20 | | 40 | | 100 | |
| | | mIoU(%) | mAcc(%) | mIoU(%) | mAcc(%) | mIoU(%) | mAcc(%) | mIoU(%) | mAcc(%) | mIoU(%) | mAcc(%) | mIoU(%) | mAcc(%) |
| DeepLabV3 | C&W (c=1) | 72.35 | 84.32 | 72.02 | 84.13 | 71.87 | 84.05 | 71.81 | 84.02 | 71.78 | 84.01 | 71.77 | 84.00 |
| | PGD | 41.81 | 64.36 | 34.5 | 59.03 | 27.61 | 54.0 | 23.73 | 50.77 | 21.47 | 48.58 | 19.84 | 47.04 |
| | SegPGD | 37.51 | 60.4 | 29.9 | 54.4 | 22.72 | 47.51 | 19.2 | 43.78 | 16.8 | 40.75 | 14.77 | 37.88 |
| | CosPGD | 36.17 | 59.41 | 27.12 | 51.6 | 18.68 | 42.8 | 14.35 | 37.02 | 12.23 | 33.71 | 10.97 | 31.3 |

### B.2.2 $l_\infty$-NORM CONSTRAINED ADVERSARIAL ATTACKS

Following, we ablate over the attack step size $\alpha$ for the $l_\infty$-norm constrained adversarial attacks and report the findings in Figure 10. We consider $\alpha \in \{0.005, 0.01, 0.02, 0.04, 0.1\}$. We can observe that the scaling in CosPGD ensures less susceptibility to the choice of step size given that it is set small enough ($\alpha \leq \epsilon$). In our work, we use step size $\alpha$=0.01 to maintain consistency with previous work (Kurakin et al., 2017; Gu et al., 2022).

### B.3 TABULAR RESULTS

Here we report the quantitative results that have already been presented in the main paper in Figures 2in tabular form. For the results reported in Figure 2, we report the results in tables 4. Here we observe that at low attack iterations (iterations=3) SegPGD implies that PSPNet is more adversarially robust than both DeepLabV3. However, after more attack iterations (iterations $\geq$ 5), SegPGD correctly implies that DeepLabV3 is more robust than PSPNet. Contrary to this, CosPGD even at low attack iterations correctly predicts DeepLabV3 to be more robust than PSPNet. This is an insight that CosPGD provides with considerably less computation.

Table 4: Comparison of performance of CosPGD to SegPGD for semantic segmentation over PASCAL VOC2012 validation dataset. We observe that CosPGD is a significantly stronger attack compared to SegPGD for both metrics and all models.

| Network | Attack method | Attack iterations | | | | | | | | | | | |
|---|---|---|---|---|---|---|---|---|---|---|---|---|---|
| | | 3 | | 5 | | 10 | | 20 | | 40 | | 100 | |
| | | mIoU(%) | mAcc(%) | mIoU(%) | mAcc(%) | mIoU(%) | mAcc(%) | mIoU(%) | mAcc(%) | mIoU(%) | mAcc(%) | mIoU(%) | mAcc(%) |
| UNet | SegPGD | 12.38 | 32.41 | 7.75 | 25.27 | 4.46 | 18.36 | 2.98 | 14.24 | 2.20 | 11.66 | 1.55 | 8.66 |
| | CosPGD | 9.67 | 29.46 | 3.71 | 15.89 | 0.61 | 3.39 | 0.06 | 0.38 | 0.03 | 0.16 | 0.01 | 0.04 |
| PSPNet | PGD | 13.79 | 31.91 | 7.59 | 21.15 | 5.44 | 16.96 | 4.48 | 14.78 | 3.80 | 13.13 | 3.72 | 13.21 |
| | SegPGD | 9.19 | 23.25 | 4.7 | 14.25 | 2.72 | 9.5 | 1.82 | 7.39 | 1.3 | 5.77 | 0.83 | 3.86 |
| | CosPGD | 7.03 | 19.73 | 2.15 | 7.6 | 0.408 | 1.44 | 0.04 | 0.11 | 0.005 | 0.021 | 0.0002 | 0.0007 |
| DeepLabV3 | PGD | 10.69 | 28.76 | 8.0 | 25.29 | 7.02 | 24.05 | 6.84 | 23.87 | 6.79 | 23.81 | 7.01 | 24.13 |
| | MI-FGSM | 10.86 | 29.39 | 7.75 | 24.97 | 6.95 | 24.06 | 6.67 | 23.52 | 6.57 | 23.48 | – | – |
| | SegPGD | 6.76 | 19.78 | 4.86 | 16.49 | 3.84 | 14.29 | 3.31 | 12.40 | 2.69 | 10.81 | 2.15 | 9.25 |
| | CosPGD | 4.44 | 14.97 | 1.84 | 7.89 | 0.69 | 3.18 | 0.12 | 0.48 | 0.08 | 0.25 | 0.005 | 0.16 |

### B.4 ADVERSARIAL TRAINING

In Figure 11 we show the segmentation masks predicted by UNet after being adversarially trained. We observe that even after 100 attack iterations, the model adversarially trained using CosPGD is making reasonable predictions. However, the model trained with SegPGD is merely predicting a blob.

In Table 5 we report the performance of models trained with various adversarial attacks against different commonly used adversarial attacks across multiple attack iterations. We observe that the

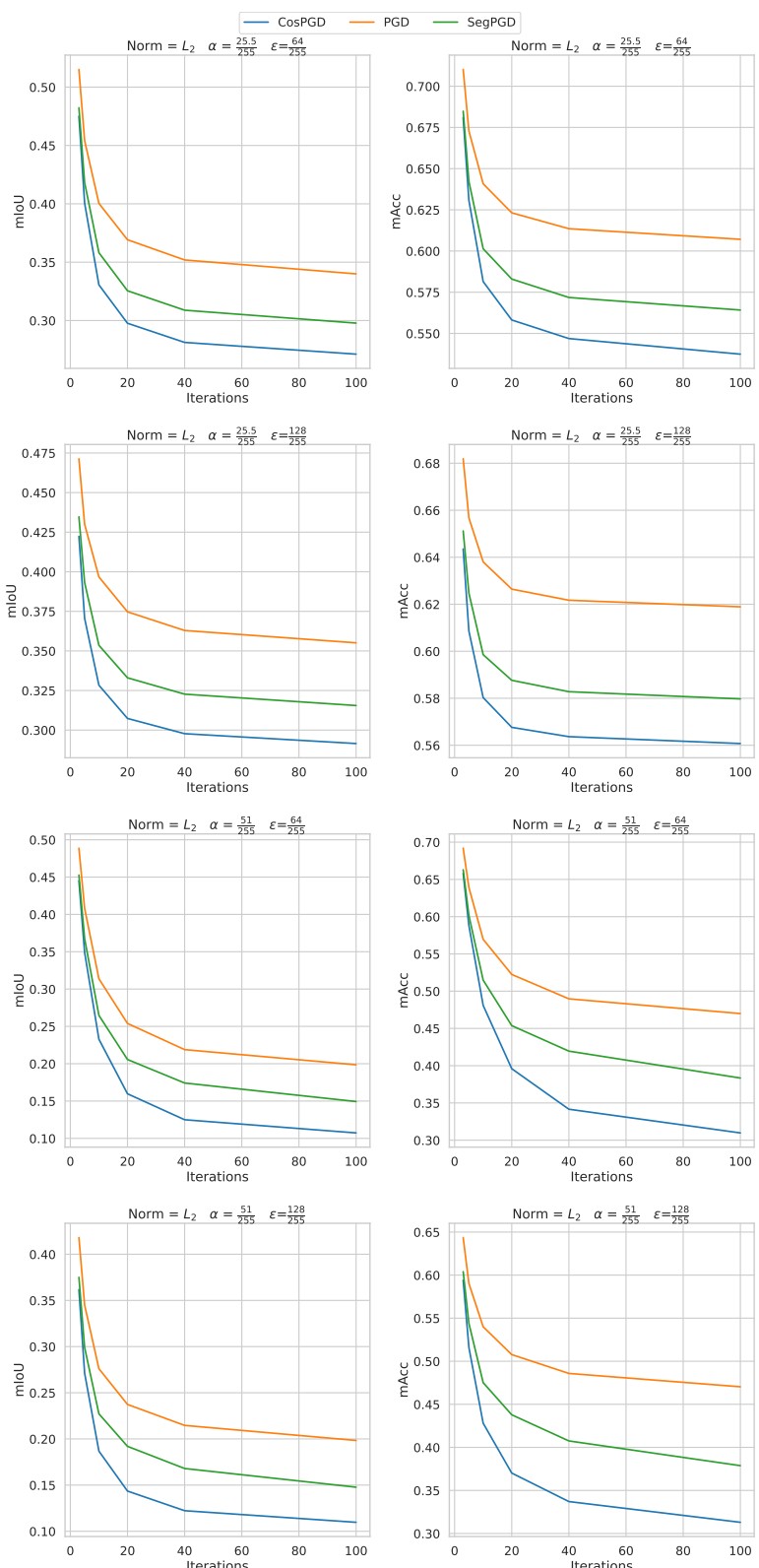

Figure 9: Comparing CosPGD to PGD and SegPGD across iterations as $l_2$-norm constrained attacks, and across $\alpha$ and $\epsilon$ values using DeepLabV3 architecture with a ResNet50 on PASCAL VOC 2012 validation dataset. Again, CosPGD outperforms previous attacks be a large margin at all attack iterations.

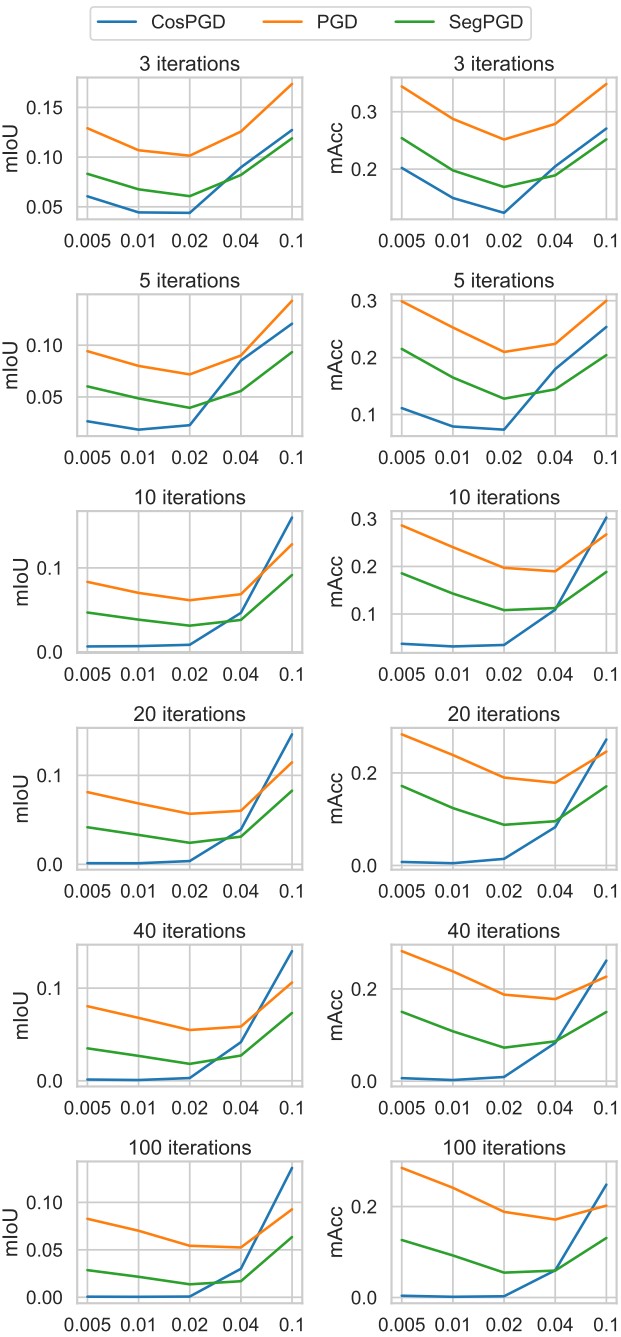

Figure 10: We ablate step sizes $\alpha$ for $l_\infty$-norm constrained CosPGD, SegPGD, and PGD attacks given different number of iterations $\in \{3, 5, 10, 20, 40, 100\}$ by attacking DeepLabV3 trained on the PASCAL VOC2012 dataset with maximal perturbation of $\epsilon = 0.03$. We can observe that the scaling in CosPGD ensures less susceptibility to the choice of step size given that it is set small enough ($\alpha \leq \epsilon$).

Table 5: Evaluating the adversarial performance of models on PASCAL VOC2012 validation dataset that are adversarially trained using PASCAL VOC2012 training dataset. "Training method" specifies the adversarial attack used during training, such that "Clean" stands for no adversarial attack being used during training. During training, 3 attack iterations were used for all adversarial attacks with $\alpha$=0.01 and $\epsilon \approx \frac{8}{255}$. These models were evaluated against multiple adversarial attacks denoted by "Attack method". We observe that models trained with CosPGD substantially outperform all the other adversarial training methods.

| Network | Training method | Attack method | Attack iterations | | | | | | | | | | | |
|---|---|---|---|---|---|---|---|---|---|---|---|---|---|---|
| | | | 3 | | 5 | | 10 | | 20 | | 40 | | 100 | |
| | | | mIoU(%) | mAcc(%) | mIoU(%) | mAcc(%) | mIoU(%) | mAcc(%) | mIoU(%) | mAcc(%) | mIoU(%) | mAcc(%) | mIoU(%) | mAcc(%) |
| UNet | | Clean | 23.18 | 46.64 | 14.58 | 35.89 | 8.21 | 24.99 | 5.57 | 18.57 | 4.14 | 14.53 | 3.6 | 11.72 |
| | PGD | PGD | 29.26 | 57.52 | 21.28 | 51.06 | 13.74 | 41.57 | 9.29 | 32.51 | 7.47 | 27.46 | 6.38 | 22.43 |
| | | SegPGD | 31.77 | 63.91 | 22.77 | 57.82 | 14.86 | 48.09 | 11.03 | 40.25 | 8.98 | 34.29 | 7.45 | 28.4 |
| | | **CosPGD** | **47.35** | **68.67** | **43.75** | **66.34** | **38.1** | **62.85** | **34.33** | **60.06** | **32.28** | **58.64** | **30.55** | **57.51** |
| | | Clean | 12.38 | 32.41 | 7.75 | 25.27 | 4.46 | 18.36 | 2.98 | 14.24 | 2.20 | 11.66 | 1.55 | 8.66 |
| | SegPGD | PGD | 29.38 | 57.82 | 21.31 | 51.35 | 13.77 | 41.72 | 9.39 | 33.15 | 7.45 | 26.98 | 6.38 | 22.26 |
| | | SegPGD | 31.69 | 63.94 | 22.47 | 57.07 | 14.82 | 47.94 | 10.9 | 40.32 | 9.09 | 34.68 | 7.33 | 27.99 |
| | | **CosPGD** | **47.16** | **68.51** | **43.85** | **66.41** | **37.64** | **62.58** | **33.99** | **59.8** | **31.91** | **58.31** | **30.48** | **57.01** |
| | | Clean | 9.67 | 29.46 | 3.71 | 15.89 | 0.61 | 3.39 | 0.06 | 0.38 | 0.03 | 0.16 | 0.01 | 0.04 |
| | CosPGD | PGD | 29.23 | 57.71 | 21.09 | 50.73 | 13.49 | 40.91 | 9.28 | 32.68 | 7.36 | 27.02 | 6.29 | 22.0 |
| | | SegPGD | 31.53 | 63.96 | 22.46 | 57.23 | 14.81 | 48.09 | 10.86 | 40.26 | 9.20 | 35.33 | 7.28 | 28.03 |
| | | **CosPGD** | **47.07** | **68.39** | **43.95** | **66.52** | **37.64** | **62.38** | **34.01** | **60.03** | **32.0** | **58.47** | **30.55** | **57.28** |
| DeepLab | | Clean | 11.02 | 30.96 | 8.50 | 27.34 | 7.63 | 26.35 | 7.57 | 26.30 | 7.59 | 26.19 | 7.39 | 25.98 |
| | PGD | PGD | 21.05 | 29.07 | 16.74 | 24.61 | 14.45 | 22.19 | 13.82 | 21.56 | 13.58 | 21.32 | 13.42 | 21.17 |
| | | SegPGD | 22.67 | 31.87 | 17.85 | 26.99 | 15.21 | 24.26 | 14.42 | 23.47 | 14.11 | 23.16 | 13.90 | 22.93 |
| | | **CosPGD** | **23.13** | **32.21** | **18.33** | **27.34** | **15.68** | **24.60** | **14.80** | **23.61** | **14.49** | **23.29** | **14.27** | **23.06** |
| | | Clean | 6.78 | 20.50 | 5.05 | 17.40 | 3.99 | 14.95 | 3.32 | 12.94 | 2.60 | 10.57 | 1.80 | 8.05 |
| | SegPGD | PGD | 20.62 | 28.54 | 16.12 | 23.79 | 13.95 | 21.42 | 13.41 | 20.84 | 13.20 | 20.61 | 13.04 | 20.42 |
| | | SegPGD | 22.06 | 31.37 | 16.89 | 26.02 | 14.27 | **23.23** | 13.57 | **22.50** | 13.33 | **22.23** | 13.09 | 21.92 |
| | | **CosPGD** | **22.33** | **31.48** | **17.15** | **26.07** | **14.54** | 23.18 | **13.89** | 22.45 | **13.67** | 22.22 | **13.54** | **22.15** |
| | | Clean | 4.71 | 16.35 | 1.94 | 8.09 | 0.61 | 3.32 | 0.24 | 1.59 | 0.09 | 0.53 | 0.08 | 0.59 |
| | CosPGD | PGD | 20.56 | 28.48 | 16.05 | 23.75 | 13.87 | 21.45 | 13.38 | 20.92 | 13.18 | 20.72 | **13.07** | 20.59 |
| | | SegPGD | 21.87 | 31.19 | 16.62 | 25.77 | 13.91 | 22.93 | 13.19 | 22.17 | 12.92 | 21.87 | 12.78 | 21.72 |
| | | CosPGD | 22.14 | 31.33 | 16.88 | 25.85 | 14.18 | 22.99 | 13.48 | 22.21 | 13.20 | 21.90 | 13.05 | 21.76 |

| Clean Image and GT Mask | Trained with CosPGD | Trained with SegPGD |
|---|---|---|

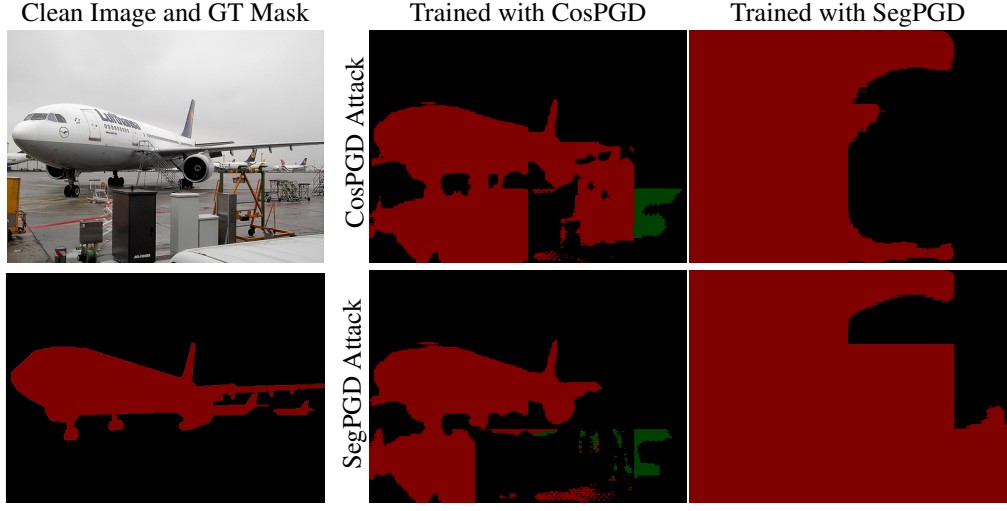

Figure 11: Predictions using UNet with ConvNeXt backbone on PASCAL VOC2012 validation dataset after 100 iterations adversarial attacks on adversarially trained models. We observe that the models adversarially trained with CosPGD are predicting reasonable masks even after 100 attack iterations, while the model trained with SegPGD is providing much worse results under both SegPGD and CosPGD attacks.

model trained with CosPGD performs the best against all considered adversarial attacks. The models were trained with 3 attack iterations of the respective "Training Method" attack during training.

In Figure 19 we present the training curves for training DeepLabV3 on the PASCAL VOC2012 training dataset using adversarial training with 50% minibatch being used for generating adversarial samples. All models are evaluated against 10 attack iterations of the respective attack.

Table 6: Comparison of performance of CosPGD to PGD as a targeted attack for optical flow estimation over KITTI15 and Sintel validation datasets using RAFT for different numbers of attack iterations. $epe$ values are compared, with respect to both, the **Target** i.e. $\overrightarrow{0}$ where a lower $epe$ indicates a better attack and Initial flow prediction (optical flow estimated by the model before any adversarial attack) where a higher $epe$ indicates a better attack. CosPGD and PGD perform similarly for a low number of iterations, where CosPGD fits the target slightly better. CosPGD significantly outperforms PGD from the 10th iteration on both metrics.

| Attack | KITTI 2015 | | | | | | MPI Sintel | | | | | | | | | | | |
| | | | | | | | | | clean | | | | | | final | | | |
| Iterations | SegPGD | | PGD | | CosPGD | | SegPGD | | PGD | | CosPGD | | SegPGD | | PGD | | CosPGD | |
| | Target↓ | Initial↑ | Target↓ | Initial↑ | Target↓ | Initial↑ | Target↓ | Initial↑ | Target↓ | Initial↑ | Target↓ | Initial↑ | Target↓ | Initial↑ | Target↓ | Initial↑ | Target↓ | Initial↑ |
| 3 | **20.57** | 11.28 | 20.7 | **11.4** | 20.6 | 11.2 | 8.35 | **6.83** | 8.3 | 6.8 | **8.1** | 6.6 | 7.58 | **7.52** | 7.6 | 7.3 | **7.5** | 7.3 |
| 5 | 14.33 | 17.75 | 14.4 | **17.8** | **14.3** | 17.7 | 6.06 | 8.97 | 6.1 | **9.0** | **5.8** | 8.8 | 5.44 | **9.43** | 5.6 | 9.4 | **5.2** | 9.3 |
| 10 | 11.08 | 21.36 | 10.5 | 22.1 | **9.0** | **23.4** | 3.51 | 11.16 | 3.4 | 11.2 | **2.9** | **11.4** | 3.13 | 11.32 | 3.1 | 11.3 | **2.6** | **11.5** |
| 20 | 7.76 | 24.55 | 8.1 | 24.6 | **6.5** | **25.8** | 2.97 | 11.61 | 2.8 | 11.7 | **2.0** | **12.1** | 2.62 | 11.7 | 2.5 | 11.8 | **1.6** | **12.1** |
| 40 | 7.53 | 24.89 | 7.3 | 25.0 | **4.8** | **27.4** | 2.66 | 11.8 | 2.8 | 11.7 | **1.6** | **12.4** | 2.4 | 11.83 | 2.6 | 12.3 | **1.3** | **12.3** |

## C  OPTICAL FLOW ESTIMATION

### C.1  TABULAR RESULTS

Here we report the results from Figure 5 comparing CosPGD to PGD as a targeted attack using RAFT for KITTI15 and Sintel datasets in tabular form in Table 6. We observe that CosPGD is more effective than PGD to change the predictions toward the targeted prediction. During a low number of iterations (iterations = 3 and 5), PGD is on par with CosPGD in increasing the $epe$ values of the predictions compared to the initial predictions on non-attacked images. However, as the number of iterations increases, CosPGD outperforms PGD for this metric as well. In the following, we report further results and compare CosPGD to a recently proposed sophisticated $l_2$-norm constrained targeted attack PCFA.

### C.2  NON-TARGETED ATTACKS FOR OPTICAL FLOW ESTIMATION

For $l_\infty$-norm constrained non-targeted attacks, CosPGD changes pixels values temperately over a larger region of the image, while PGD changes it drastically but only for a small region in the image. This can be observed in Figure 12 when CosPGD and PGD are compared as $l_\infty$-norm constrained non-targeted attacks for optical flow estimation. We observe that both CosPGD and PGD are performing at par as both have very similar $epe$ values across iterations. However, CosPGD across iterations has a lower $epe$-$f1$-$all$ value. As shown by Equation 7 in Section A.1.2, $epe$-$f1$-$all$ is the measure of average overall $epe$ values that are above a modest threshold. Therefore, both CosPGD and PGD have very similar $epe$ scores while CosPGD has a significantly lower $epe$-$f1$-$all$ compared to PGD. This implies that CosPGD and PGD are performing at par, however, PGD is drastically changing $epe$ values at certain pixels, while CosPGD is changing $epe$ values temperately over considerably more pixels. Figure 13 shows this qualitatively for 4 randomly chosen samples.

### C.3  COMPARISON TO PCFA

Further, we compare CosPGD as a $l_2$-norm constrained targeted attack to the recently proposed *state-of-the-art* $l_2$-norm constrained targeted attack PCFA (Schmalfuss et al., 2022). For comparison. we use the same settings as those used by the authors for both attacks, for 20 attack iterations (steps), generating adversarial patches for each image individually, bounded under the change of variables methods proposed by Schmalfuss et al. (2022). Here, we observe that a sophisticated $l_2$-norm constrained targeted attack, PCFA that does not utilise pixel-wise information for generating adversarial patches over all considered networks and datasets, performs similar to CosPGD. We compare over the performance over RAFT, PWCNet (Sun et al., 2018), GMA (Jiang et al., 2021) and SpyNet (Ranjan and Black, 2017) We consider both targeted settings proposed by Schmalfuss et al. (2022), i.e. target being a zero vector $\overrightarrow{0}$ and target being the negative of the initial prediction (*negative flow*). We compare the average $epe$ over all images. A lower $AEE$ is w.r.t. Target and higher $AEE$ w.r.t. initial indicate a stronger attack. In Table 8(currently included at the end of the appendix to not disturb the table numbers), we compare PCFA and CosPGD on multiple datasets, multiple networks over 3 random seeds.

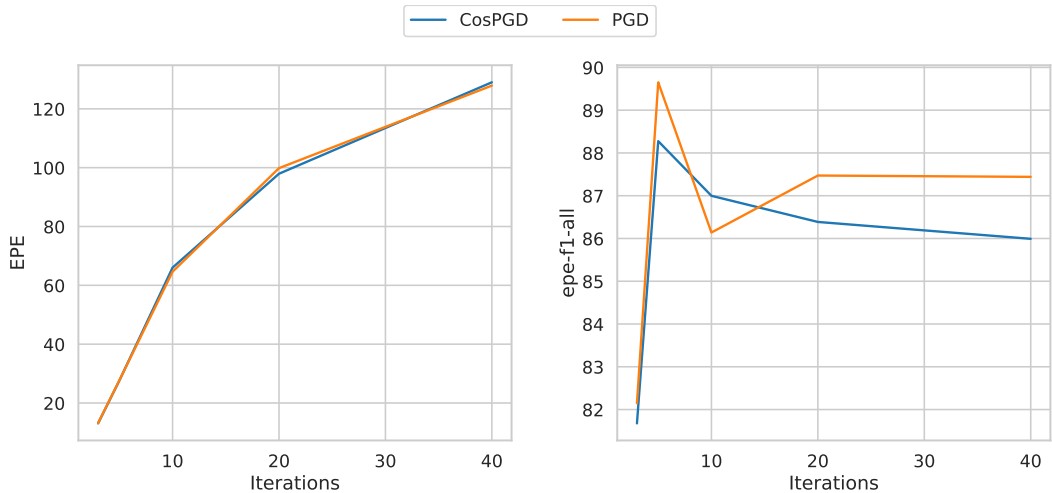

Figure 12: Comparing CosPGD and PGD as $l_\infty$-norm constrained non-targeted attacks for optical flow estimation using RAFT on KITTI 2015 validation dataset.

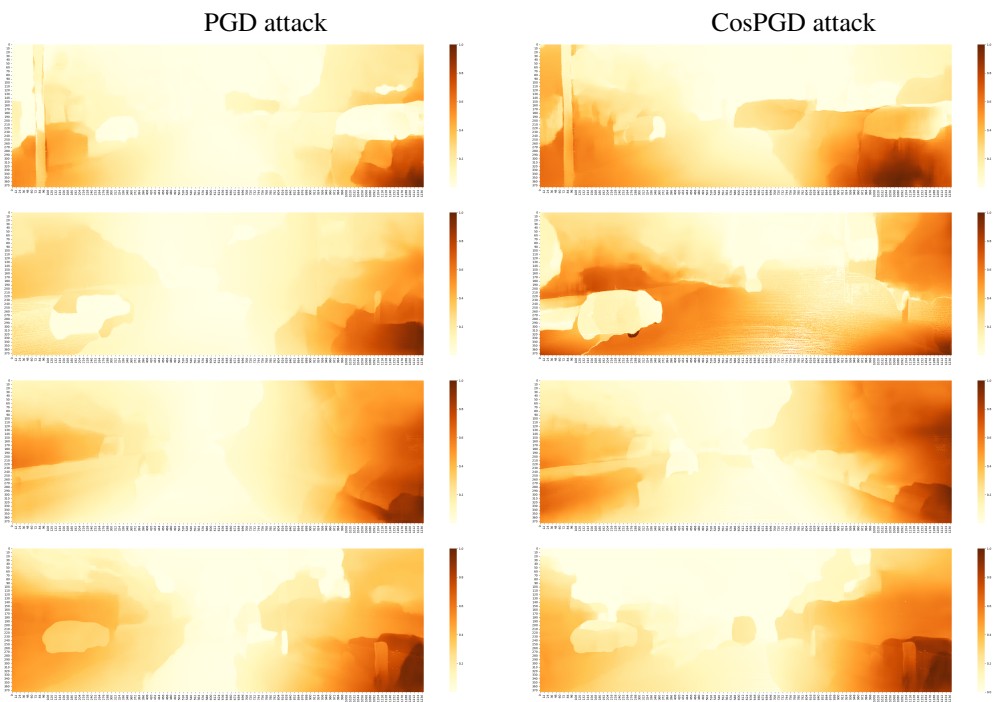

Figure 13: Comparing change in pixel-wise $epe$ values w.r.t. initial $epe$ values after 40 iterations of PGD and CosPGD as non-targeted $l_\infty$-norm constrained attacks on RAFT using KITTI15 validation set. The values for each image are: $\frac{|epe_{adv} - epe_{initial}|}{max(epe_{adv})}$ where $epe_{adv}$ & $epe_{initial}$ are pixel-wise $epe$ values of the final adversarial sample and the initial non-attacked image, respectively.

Figure 14, provides an overview of the comparison between the two methods, using targets as $\overrightarrow{0}$ and *negative flow*. Figures 15, 16, provide further details compares both methods when using $\overrightarrow{0}$ and *negative flow* as the target, respectively.

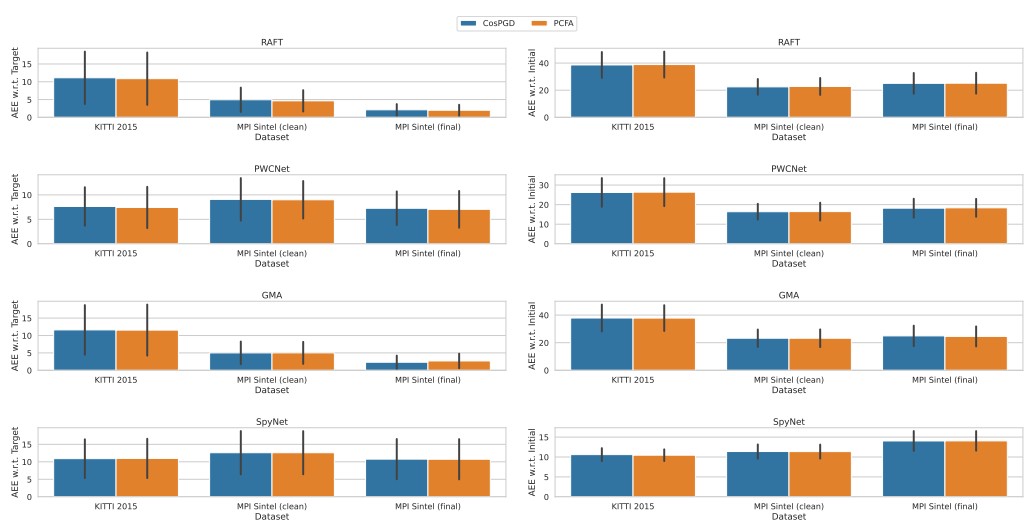

AEE w.r.t. Target, lower is better          AEE w.r.t. Initial, higher is better

Figure 14: Comparison of mean and standard deviation of the results using different targets, $\overrightarrow{0}$ and *negative flow* for CosPGD and PCFA. A lower $AEE$ is w.r.t. Target and a higher $AEE$ w.r.t. initial indicate a stronger attack.

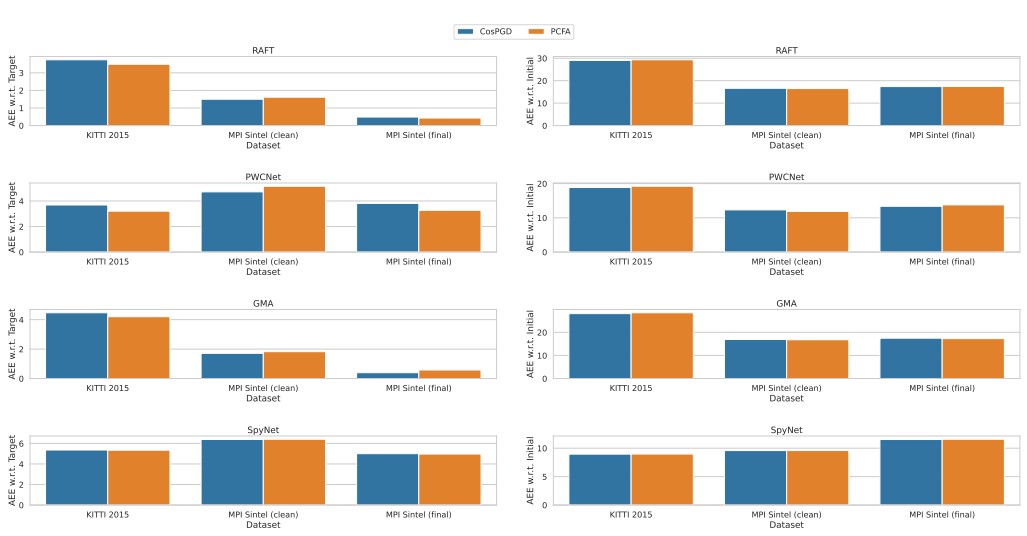

AEE w.r.t. Target, lower is better          AEE w.r.t. Initial, higher is better

Figure 15: Comparison of PCFA and CosPGD when using $\overrightarrow{0}$ as the target. A lower $AEE$ is w.r.t. Target and a higher $AEE$ w.r.t. initial indicate a stronger attack.

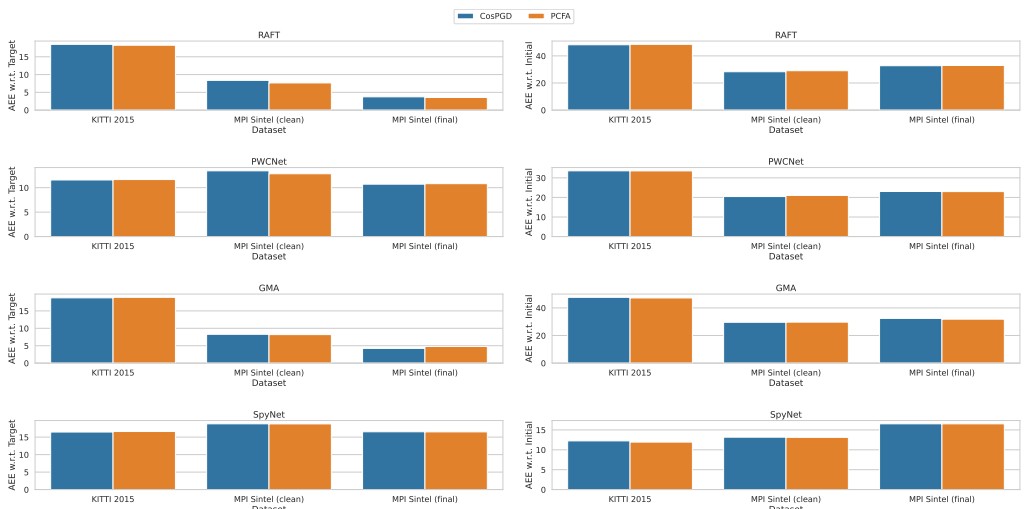

AEE w.r.t. Target, lower is better          AEE w.r.t. Initial, higher is better

Figure 16: Comparison of PCFA and CosPGD when using *negative flow* as the target. A lower $AEE$ is w.r.t. Target and a higher $AEE$ w.r.t. initial indicate a stronger attack.

## D  IMAGE RESTORATION TASKS

Following, we provide further results and discussion on the two considered image restoration tasks namely, Image Deblurring in Section D.1 and Image Denoising in Section D.2

### D.1  IMAGE DEBLURRING MODELS

In Figure 17 for the Baseline network, we observe that both CosPGD and PGD are performing at par. While for the newly proposed NAFNet, PGD is still estimating NAFNet's adversarial robustness to be very similar to the Baseline network and only after 20 attack iterations it is estimating correctly that NAFNet is not as robust as the Baseline network. However, CosPGD reveals that NAFNet is not as robust as the baseline even at a low number of iterations (3 attack iterations). This valuable insight regarding model robustness of newly proposed transformer-based image restoration models is provided by CosPGD with considerably less computation.

Following the discussion from Section 4.3, in Figure 7 for the Baseline network we also observe that SegPGD here is significantly weaker due to its limitation to image classification tasks as discussed in Section 3. However, for NAFNet, from 5 attack iterations onwards SegPGD is outperforming PGD, while still being weaker than CosPGD. This, interesting improvement in the performance of SegPGD as an adversarial attack can be attributed to the pixel-wise nature of the attack, similar to CosPGD further highlighting the benefits of utilizing pixel-wise information when crafting adversarial attacks for pixel-wise prediction tasks.

Additionally, we report the findings on many recently proposed state-of-the-art image restoration models using CosPGD in Table 7.

### D.2  NON-TARGETED ATTACKS FOR IMAGE DENOISING TASK

**Dataset.**  For the image denoising task, following work from Chen et al. (2022); Zamir et al. (2022) we use the Smartphone Image Denoising Dataset (SSID) (Abdelhamed et al., 2018). This dataset consists of 160 noisy images taken from 5 different smartphones and their corresponding high-quality ground truth images. Similar to the image deblurring task, we report the $PSNR$ and $SSIM$ values as metrics for this image restoration task as well.

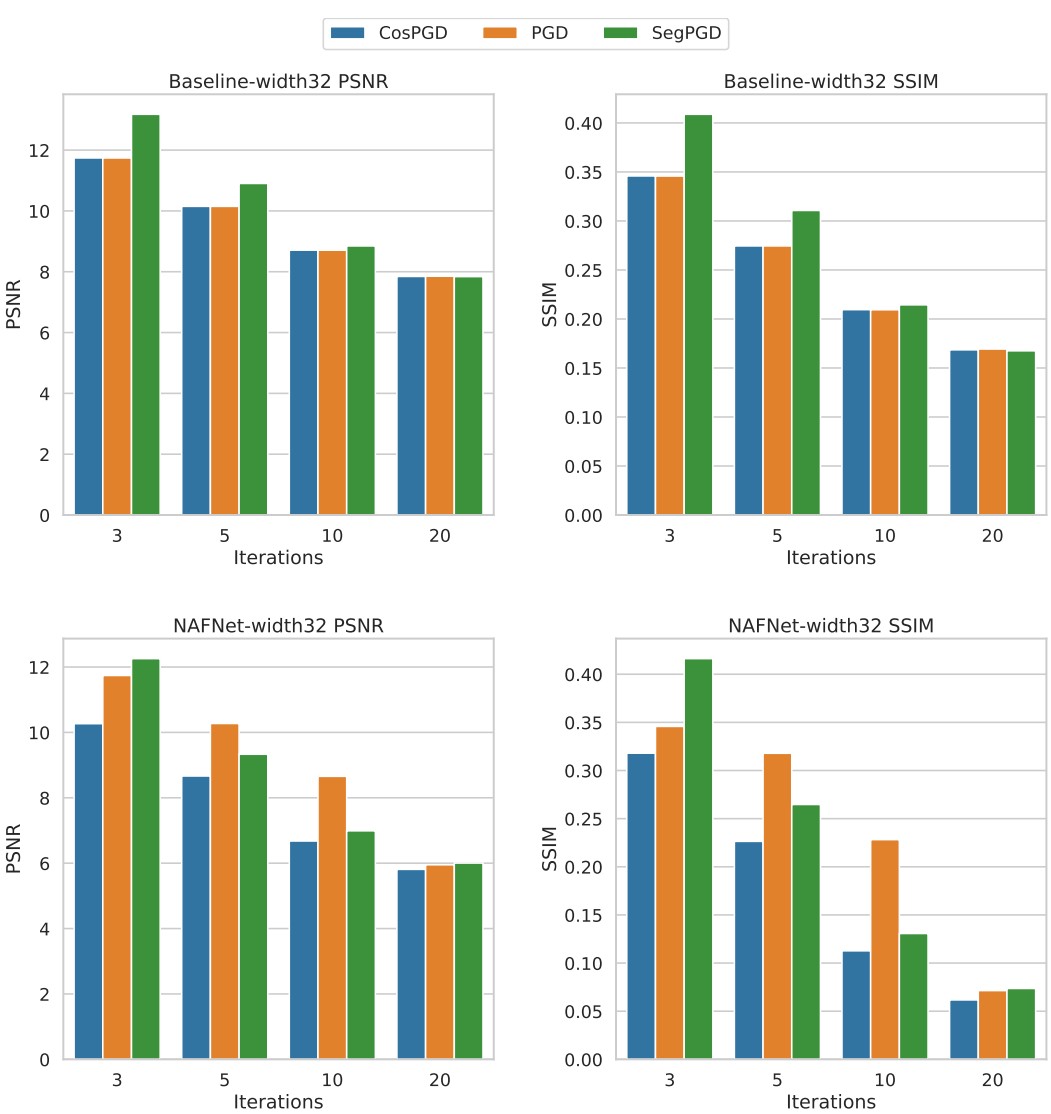

Figure 17: Non-targeted $l_\infty$-norm constrained CosPGD, PGD, and SegPGD attacks on the "Baseline network" and NAFNet for image deblurring task on the GoPro dataset, recently proposed by Chen et al. (2022) as the state-of-the-art networks for image restoration tasks. The "Baseline network" is significantly more robust than the NAFNet and thus the performance of the Baseline network against CosPGD attack is at par with its performance against PGD. However, PGD indicates at low attack iterations (iterations $\leq 10$) that NAFNet is more robust than "Baseline network" and only after 20 attack iterations its correctly indicates that NAFNet is less robust. However, CosPGD is able to draw this conclusion at merely 3 attack iterations.

Table 7: Comparison of clean and adversarial performance of image reconstruction models, as considered by Agnihotri et al. (2023). '+ADV' denotes FGSM adversarial training with a 50-50 mini-batch split for generating an adversarial sample.

| Architecture | Clean | | CosPGD | | | | | | PGD | | | | | |
|---|---|---|---|---|---|---|---|---|---|---|---|---|---|---|
| | | | 5 attack itrs | | 10 attack itrs | | 20 attack itrs | | 5 attack itrs | | 10 attack itrs | | 20 attack itrs | |
| | PSNR | SSIM | PSNR | SSIM | PSNR | SSIM | PSNR | SSIM | PSNR | SSIM | PSNR | SSIM | PSNR | SSIM |
| **Restormer**(Zamir et al., 2022) | 31.99 | 0.9635 | **11.36** | **0.3236** | 9.05 | 0.2242 | 7.59 | 0.1548 | 11.41 | 0.3256 | 9.04 | 0.2234 | 7.58 | 0.1543 |
| + **ADV** | 30.25 | 0.9453 | 24.49 | 0.81 | **23.48** | **0.78** | 21.58 | 0.7317 | **24.5** | 0.8079 | **23.5** | **0.7815** | 21.58 | 0.7315 |
| Baseline(Chen et al., 2022) | 32.48 | 0.9575 | 10.15 | 0.2745 | 8.71 | 0.2095 | 7.85 | 0.1685 | 10.15 | 0.2745 | 8.71 | 0.2094 | 7.85 | 0.1693 |
| + ADV | 30.37 | 0.9355 | 15.47 | 0.5216 | 13.75 | 0.4593 | 12.25 | 0.4032 | 15.47 | 0.5215 | 13.75 | 0.4592 | 12.24 | 0.4026 |
| NAFNet(Chen et al., 2022) | 32.87 | 0.9606 | 8.67 | 0.2264 | 6.68 | 0.1127 | 5.81 | 0.0617 | 10.27 | 0.3179 | 8.66 | 0.2282 | 5.95 | 0.0714 |
| + ADV | 29.91 | 0.9291 | 17.33 | 0.6046 | 14.68 | 0.509 | 12.30 | 0.4046 | 15.76 | 0.5228 | 13.91 | 0.4445 | 12.73 | 0.3859 |

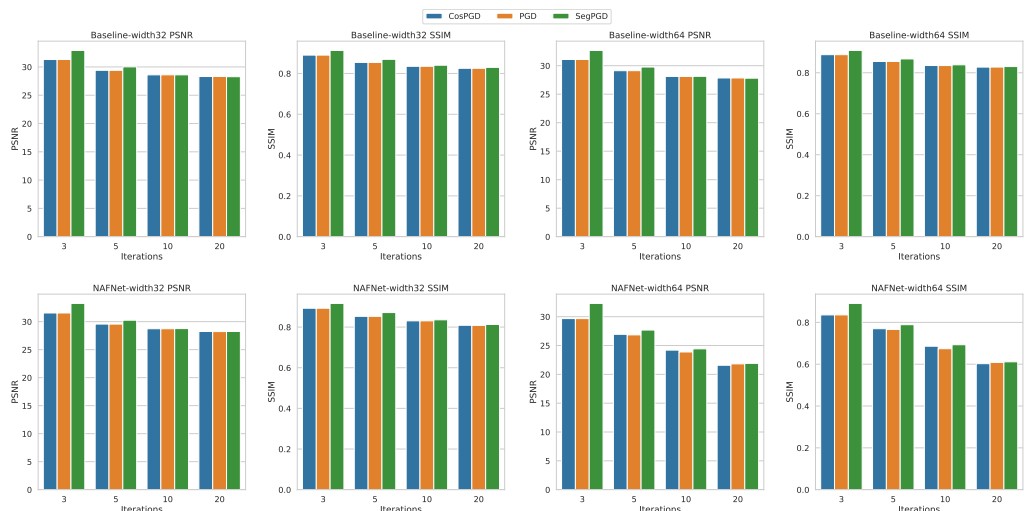

Figure 18: Comparing CosPGD to PGD and SegPGD as $l_\infty$-norm constrained non-targeted attacks for the image denoising task using Baseline network (top row) and NAFNet (bottom row) on SSID dataset. A lower value of PSNR and SSIM indicate a stronger attack.

**Discussion.** Further extending the findings from Section C.2 we report $l_\infty$-norm constrained non-targeted attacks for the image denoising on the SSID dataset using the Baseline network and NAFNet (as proposed by Chen et al. (2022)) in Figure. 18. We observe that both CosPGD and PGD are performing at par for both, the Baseline network and NAFNet. Additionally, similar to findings in Section 4.3, SegPGD is unable to perform at par with CosPGD and PGD.

After both CosPGD and PGD attacks it appears that the image denoising networks are relatively more robust than image deblurring networks. These findings also correlate with Xie et al. (2019), as they report that feature denonising improves model robustness against adversarial attacks.

## E  FURTHER DISCUSSION ON LIMITATIONS OF COSPGD

Similar to most white-box adversarial attacks (Goodfellow et al., 2014; Kurakin et al., 2017; Madry et al., 2017; Wong et al., 2020b; Gu et al., 2022), CosPGD currently requires access to the model's gradients for generating adversarial examples. While this is beneficial for generating adversaries, it limits the applications of the non-targeted settings as many benchmark datasets (Menze and Geiger, 2015; Butler et al., 2012; Wulff et al., 2012; Everingham et al., 2012) do not provide the ground truth for test data. Evaluations of the validation datasets certainly show the merit of the attack method. CosPGD mitigates this limitation by also being applicable as an effective targeted attack. Nevertheless, it would be interesting to study the attack on test images as well in an untargeted setting, due to the potential slight distribution shifts pre-existing in the test data. While CosPGD is significantly more efficient than other existing adversarial attacks, all white-box adversarial attacks

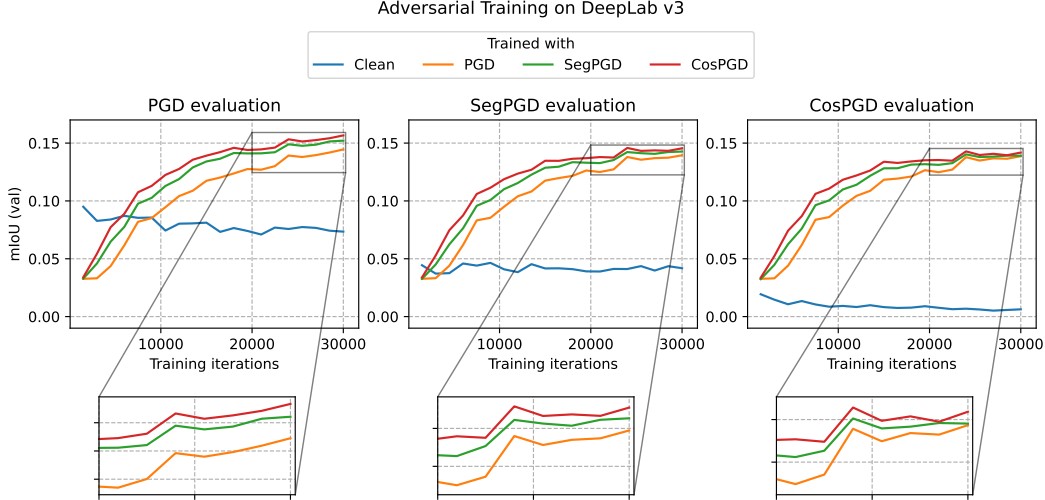

Figure 19: DeepLabV3 adversarially trained using different adversarial attacks for 3 iterations during training using 50% of the minibatch for generating adversarial samples. All checkpoints are evaluated against 10 attack iterations of the respective attacks. We observe that the model trained with CosPGD outperforms all other adversarial training methods considered against all attacks.

are time and memory consuming and benchmarking them across multiple downstream tasks, datasets, and networks is a very time-consuming process.

Additionally, as discussed in Section 5 paragraph Limitations, there would exist settings where approaches like pixel-wise PGD work at par with CosPGD as the *epe* can be changed equally by changing all pixel-wise regressing estimates slightly or changing only a few of them drastically, as can also be seen in the results in C.2.

## F    ADVERSARIAL TRAINING CURVES FOR DEEPLABV3

In Figure 19 we present the training curves for training DeepLabV3 on the PASCAL VOC2012 training dataset using adversarial training with 50% minibatch being used for generating adversarial samples.

Upon acceptance, we would include Figure 19 in Section B.4. It has been included here, for now, to not disturb the figure numbering.

## G    EXTRA REBUTTAL RESULTS

Table 8: Comparison of performance of CosPGD to PCFA as a targeted $l_2$-norm constrained attack for optical flow estimation over KITTI2015 and Sintel validation datasets using different optical flow models over 3 random seeds. Average $epe$ values are compared, with respect to both, the **Target** where a lower $epe$ indicates a better attack and **Initial flow prediction** (optical flow estimated by the model before any adversarial attack) where a higher $epe$ indicates a better attack. We compare over both targets used by Schmalfuss et al. (2022), i.e. zero vector $\vec{0}$ and Negative of the Initial Flow. **CosPGD and PCFA performance is very comparable.** This table will be included in Section C.3 upon acceptance.

| | KITTI 2015 | | | | | | | |
| | Target $\vec{0}$ | | | | Negative Initial Flow | | | |
| **Model** | AEE wrt Target↓ | | AEE wrt Initial↑ | | AEE wrt Target↓ | | AEE wrt Initial↑ | |
| | CosPGD | PCFA | CosPGD | PCFA | CosPGD | PCFA | CosPGD | PCFA |
|---|---|---|---|---|---|---|---|---|
| GMA | $28.69 \pm 0.12$ | $28.67 \pm 0.17$ | $3.89 \pm 0.09$ | $3.89 \pm 0.15$ | $47.00 \pm 0.40$ | $47.08 \pm 0.69$ | $19.22 \pm 0.53$ | $19.20 \pm 0.57$ |
| PWCNet | $19.13 \pm 0.04$ | $18.96 \pm 0.08$ | $3.25 \pm 0.08$ | $3.47 \pm 0.14$ | $33.13 \pm 0.25$ | $33.13 \pm 0.26$ | $12.01 \pm 0.20$ | $12.02 \pm 0.22$ |
| RAFT | $29.09 \pm 0.03$ | $29.17 \pm 0.11$ | $3.75 \pm 0.05$ | $3.63 \pm 0.10$ | $48.83 \pm 0.35$ | $48.93 \pm 0.29$ | $17.97 \pm 0.29$ | $17.81 \pm 0.27$ |
| SpyNet | $9.00 \pm 0.01$ | $9.01 \pm 0.03$ | $5.31 \pm 0.01$ | $5.35 \pm 0.06$ | $12.10 \pm 0.02$ | $12.08 \pm 0.05$ | $16.47 \pm 0.03$ | $16.44 \pm 0.05$ |
| | **MPI Sintel (clean)** | | | | | | | |
| GMA | $16.87 \pm 0.14$ | $16.76 \pm 0.11$ | $1.75 \pm 0.15$ | $1.85 \pm 0.10$ | $29.25 \pm 0.38$ | $29.05 \pm 0.38$ | $8.58 \pm 0.34$ | $8.82 \pm 0.37$ |
| PWCNet | $12.20 \pm 0.21$ | $12.18 \pm 0.07$ | $4.87 \pm 0.17$ | $4.75 \pm 0.12$ | $20.57 \pm 0.21$ | $20.43 \pm 0.21$ | $13.20 \pm 0.13$ | $13.21 \pm 0.29$ |
| RAFT | $16.42 \pm 0.03$ | $16.46 \pm 0.05$ | $1.69 \pm 0.04$ | $1.65 \pm 0.06$ | $29.01 \pm 0.11$ | $29.20 \pm 0.01$ | $7.67 \pm 0.11$ | $7.47 \pm 0.05$ |
| SpyNet | $9.69 \pm 0.01$ | $9.75 \pm 0.07$ | $6.40 \pm 0.05$ | $6.35 \pm 0.00$ | $13.08 \pm 0.01$ | $13.17 \pm 0.03$ | $18.75 \pm 0.02$ | $18.76 \pm 0.06$ |
| | **MPI Sintel (final)** | | | | | | | |
| GMA | $17.34 \pm 0.07$ | $17.31 \pm 0.11$ | $0.53 \pm 0.07$ | $0.54 \pm 0.11$ | $32.11 \pm 0.20$ | $32.04 \pm 0.24$ | $4.57 \pm 0.22$ | $4.64 \pm 0.24$ |
| PWCNet | $13.61 \pm 0.10$ | $13.44 \pm 0.14$ | $3.52 \pm 0.13$ | $3.66 \pm 0.12$ | $23.00 \pm 0.30$ | $23.01 \pm 0.06$ | $10.84 \pm 0.28$ | $10.75 \pm 0.05$ |
| RAFT | $17.38 \pm 0.04$ | $17.36 \pm 0.03$ | $0.55 \pm 0.09$ | $0.50 \pm 0.03$ | $32.72 \pm 0.22$ | $32.72 \pm 0.14$ | $3.71 \pm 0.21$ | $3.75 \pm 0.13$ |
| SpyNet | $11.56 \pm 0.01$ | $11.59 \pm 0.03$ | $4.97 \pm 0.01$ | $4.97 \pm 0.01$ | $16.51 \pm 0.01$ | $16.55 \pm 0.06$ | $16.52 \pm 0.01$ | $16.47 \pm 0.05$ |

