# OpenReview forum: "CosPGD: an efficient white-box adversarial attack for pixel-wise prediction tasks"
_ICLR.cc/2024/Conference — Submitted to ICLR 2024_

### Official Review · Reviewer_3VUb · 2023-10-27

**Soundness:** 3 good
**Presentation:** 3 good
**Contribution:** 2 fair
**Rating:** 5
**Confidence:** 4

**Summary:**

This paper presents a white-box adversarial attack CosPGD for dense predictions tasks such as semantic segmentation, optical flow and image restoration. CosPGD adopts the cossine similarity to weight the basic PGD attack, which has better interpretability compared to the weight adjustment based on the number of iterations used in SegPGD. Experimental results show CosPGD is strong attack performance in multi tasks.

**Strengths:**

1. The authors discuss the differences and advantages of PGD and SegPGD.
2. Compared to SegPGD, CosPGD has a broader generality, which can be applied not only to pixel classification tasks but also to pixel regression tasks.

**Weaknesses:**

1. The core of the proposed method is very similar to SegPGD, as both aim to focus on the pixels where the attack has not been successful yet (e.g. pixels with large cosine similarity weight). Therefore, the novelty is limited.
2. Ablation experiments lacking other metrics like cosine distance.
3. Lack of performance comparison experiments with state-of-the-art methods [1] for semantic segmentation tasks.

[1] Rony J, Pesquet J C, Ben Ayed I. Proximal Splitting Adversarial Attack for Semantic Segmentation[C]//Proceedings of the IEEE/CVF Conference on Computer Vision and Pattern Recognition. 2023: 20524-20533.

**Questions:**

See Weakness.

---

> ### Author Response · Authors · 2023-11-15
> **Authors' rebuttal**
>
> Dear Reviewer 3VUb,
>
> Thank you very much for your insightful review. We hope to answer your questions in a satisfactory manner.
>
> 1. The proposed CosPGD is motivated by SegPGD and indeed the equations look similar. Yet, there are significant conceptual differences that provide a strong benefit of CosPGD over SegPGD, which we also discuss on page 3 in our submission. Please refer to our general reply for details on the conceptual differences between the two and the resulting benefit of CosPGD. Please note that no other existing attack scales the loss pixel-wise using **similarity between the posterior and target distributions**. All attacks focus on final model predictions. This makes CosPGD significantly novel. Does this answer the important concern?
>
>
> 2. To the best of our knowledge, cosine distance is not the most commonly evaluated metric for any of the tasks considered: semantic segmentation, optical flow estimation, and image restoration(we evaluate structural similarity for this task).
> For all our considered models and tasks, we report all the metrics that are used by the authors to evaluate their proposed models and attacks.
>
>
> 3. Our experiments focus on the wide applicability of CosPGD rather than on outperforming a particular, dedicated approach on a particular task. We will try to provide results comparing our method to [1] for the final version, which is however not trivial since [1] proposes a minimal attack and CosPGD is an epsilon-bounded attack. In any case, we now cite in the revised version and will discuss this paper.
> **Please note** that according to the ICLR guidelines, https://iclr.cc/Conferences/2024/ReviewerGuide, publications from recent conferences (“published [...] within the last four months”: “on or after May 28, 2023”) are assumed to be contemporaneous work and a comparison is not required - this is the case for [1].
>
> We hope we were able to answer all your questions to your satisfaction. Please let us know if you have further questions or concerns.
>
> Best Regards
>
> Authors of Paper #909

---

> ### Author Response · Authors · 2023-11-20
> **Gentle Reminder**
>
> Dear Reviewer 3VUb,
>
> We were curious if you happened to find the time to read our official comments and incorporated changes in the revised version. If so, we would be glad to answer any further questions or doubts you might have.
>
> Best Regards,
>
> Authors of Paper #909

---

> > ### Comment · Reviewer_3VUb · 2023-11-22
> >
> > For Q1, we believe that the most basic cross-entropy is also a measure between distributions. Therefore, we have the same idea as Reviewer YMdT and think it is a simple improvement of SegPGD. In addition, the current version does not explain why this is better, visually or quantitatively.
> >
> > For Q2, our question is why does it have to be cos distance? Is KL or JS distance also possible?
> >
> > Looking forward to your further explanation.

---

> ### Author Response · Authors · 2023-11-22
>
> Thank you for rephrasing your question! We think we better understand the question Q1 now, from a mathematical perspective.
>
> Both SegPGD and CosPGD propose to modify the update step of the original PGD applied to each pixel, as given in our equation (1).
>
> $sign\nabla_{\boldsymbol{X}}L$
>
>  Here, we write L to denote the original model loss with respect to the current model $f$ 's prediction based on an adversarial sample
> $\boldsymbol X$
>
> and the one-hot encoded ground truth
> $Y$.
>
> For segmentation, i.e. in SegPGD $L$ is a cross-entropy loss as specified in their equation (2) . In our paper, we also use a cross entropy loss when considering segmentation, but $L$ can in principle be any loss of the respective original model, as in PGD. For optical flow, we use an optical flow loss, to be consistent with PGD.
>
> In SegPGD, the above update is modified to
>
> $sign\nabla_{\boldsymbol{X}}(\frac{1-\lambda}{N}\sum_{i\in P^T} L_i  + \frac{\lambda}{N}\sum_{k\in P^F} L_k)$
>
> where  $P^T$
> is the set of correctly classified pixels and  $P^F$
> is the set of wrongly classified pixels, $N$ is the total number of pixels and $\lambda$ is a scaling factor between the two parts of the loss that is set heuristically. See their equation (4) for details.
>
> For positive $\lambda$, this equation could be rewritten as
>
> $sign\nabla_{\boldsymbol{X}}(\frac{1}{N}\sum_{i\in P^T\cup P^F} (1- |\lambda - |(argmax(f(\boldsymbol{X}))-Y|/2|) L_i) $
>
> for adversarial examples
> $\boldsymbol{X}$
>
> i.e. $|\lambda - |(argmax(f(\boldsymbol{X}))-Y|/2|$ equals $1-\lambda$ for incorrect predictions, it equals $\lambda$ for correct predictions.
>
> You can consider this representation of SegPGD to be the starting point of our argument: no matter what loss to use for L, we argue that the weighting of the pixel-wise loss with this weight after the argmax is an issue: it limits SegPGD to applications where the correctness of the prediction can be evaluated in a binary way, and it disregards the actual prediction scores. This is why CosPGD proposes in our equation (5) proposes to use a continuous measure of correctness instead:
>
> $sign\nabla_{\boldsymbol{X}}(\frac{1}{N}\sum_{i\in P^T\cup P^F} cos(softmax(f(\boldsymbol{X})), Y) L_i) $
>
> Please note that this facilitates CosPGD to operate on the segmentation scores instead of the final argmax predictions. Positive side aspects are that we do not need to set any heuristic parameter $\lambda$ and that we can directly apply this same procedure for a wide variety of tasks beyond segmentation. The experiments we show demonstrate the significant benefit of considering the actual prediction scores w.r.t. the cosine similarity of their softmax to the ground truth.
> We empirically show the benefit of CosPGD over SegPGD in a wide variety of experiments.
>
> Does this address your concern with respect to the difference of CosPGD to SegPGD?

---

> ### Author Response · Authors · 2023-11-22
> **Q2**
>
> Regarding question Q2, we would like to reconsider the example of semantic segmentation, i.e. the formulation that you see above, where both SegPGD and CosPGD scale the cross-entropy loss per pixel.
> Then, in CosPGD, we have the softmax of the prediction scores which means that the input to the cosine similarity is a set of two vectors, both of which have values between zero and one and are normalized to one. The one-hot encoded label vector $Y$ has values that are either zero or one.
> This means that the ground truth $Y$ is a vector of binary values and just indicates the correct label. Therefore, we did not consider distribution losses (KL or JS) in general. This might be an interesting future research direction.
>
> In principle, for the proposed scaling of the pixel-wise loss to be reasonable, the scaling value for the loss at each location also needs to be between zero and one.
> We acknowledge that one could consider further variants for the scaling. Cross-entropy between the prediction and the GT will however map to an inappropriate value range, similar to  L2 or L1 distances between these two vectors, which would need to be normalized. The cosine similarity has the desired properties and seems appropriate to assess the angle between the softmax prediction and the ground truth. Therefore, the cosine similarity is an intuitive choice. We will add a respective discussion to the revision.

---

### Official Review · Reviewer_E2Ss · 2023-10-27

**Soundness:** 2 fair
**Presentation:** 3 good
**Contribution:** 2 fair
**Rating:** 5
**Confidence:** 4

**Summary:**

The paper proposed a white-box adversarial attack method CosPGD that considers the cosine similarity between predictions and targets for each pixel.  The authors claimed that CosPGD can be used for various pixel-wise prediction tasks, outperforming existing attacks on semantic segmentation and providing insights into model performance. It is similar to SegPGD and the experiments are insufficient to validate the advantage of the proposed method.

**Strengths:**

1) The authors introduce the principle and method of CosPGD clearly in **Sec.3**.

2) The universal design for loss function of CosPGD make it be applicable to a wide range of pixel-wise prediction tasks.

**Weaknesses:**

1) Take the non-targeted attack as an example, the proposed loss function in Eq.(5) $L_{\mathrm{cos}}=\frac{1}{H \times W} \sum_{H \times W} \cos (\overrightarrow{\text { pred }}, \overrightarrow{\text { target }}) \cdot L\left(f_{\text {net }}\left(\boldsymbol{X}^{\text {adv } v}\right), \boldsymbol{Y}\right),$
   is very similar with the loss function of SegPGD[1]  $L_{SegPGD} = \frac{1}{{H}\times{W}} \sum_{j\in P^T} L_j + \frac{1}{{H}\times{W}} \sum_{k\in P^F} L_k$.  Thus, the novelty is limited.

2) Although it claims that CosPGD can be used for various pixel-wise prediction tasks, but it does not bring about significant improvement compared to SegPGD[1] in image restoration task as shown in **Fig.7**, especially with 20 times iterations.

3） In **Sec.4.2** the paper identify their method perform in optical flow task, but it only did experiments compared with PGD[2] in **Fig.5**. I wonder how is the SegPGD[1] perform in optical flow task?

4） In **Sec4.3** the authors said "We observe that at low number of attack iterations (3 attack iterations) it performs significantly worse than PGD, thus demonstrating its limitation on this task." However, the SegPGD[1] is need to adjust their balance factor during the attack iteration, and as far as I know, white box attacks usually don't compare attack performance at low iterations. So I do not think it is fair to compare with SegPGD[1] in 3 attack iterations.

Ref. [1] Gu J, Zhao H, Tresp V, et al. Segpgd: An effective and efficient adversarial attack for evaluating and boosting segmentation robustness[C]//European Conference on Computer Vision. Cham: Springer Nature Switzerland, 2022: 308-325.

Ref. [2] A. Madry, A. Makelov, L. Schmidt, et al. Towards deep learning models resistant to adversarial attacks[J]. arXiv preprint arXiv:1706.06083, 2017.

-----------------------------------
After Rebuttal
------------------------------------
Thanks a lot for the authors' rebuttal. The main concern still lies in its novelty.
 (1) While the rebuttal claims that "there are no other attack method scaling the loss pixel-wise using similarity between the posterior and target distributions", which seems a bit trivial and cannot be regarded as a main contribution to this field.
(2) Although SegPGD can not be directly applied to image restoration, it can be adapted to other supervised learning tasks by doing some simple modifications.
(3) Considering that authors have provided lots of experiments and analysis, I have upgraded the score.

**Questions:**

Please see the weakness.

---

> ### Author Response · Authors · 2023-11-15
> **Authors' rebuttal**
>
> Dear Reviewer E2Ss,
>
>
> Thank you very much for your insightful review. We hope to answer your questions in a satisfactory manner.
>
> 1. The proposed CosPGD is motivated by SegPGD and indeed the equations might look similar at first glance. Yet, there are significant conceptual differences that provide a strong benefit of CosPGD over SegPGD, which we also discuss on page 3 in our submission. Please refer to our general reply for details on the conceptual differences between the two and the resulting benefit of CosPGD. Please note, that no other existing attack scales the loss pixel-wise using **similarity between the posterior and target distributions**. All attacks focus on final model predictions. This makes CosPGD significantly novel. Does this answer this important concern?
>
>
> 2. For image restoration, SegPGD can not be directly applied but by defining a threshold to determine from when on a prediction is considered to be correct. For the comparison on image restoration, we assume that the prediction should be as correct as possible, i.e. assume the lowest numerically possible threshold. After this adaptation, SegPGD’s performance comes close to the performance of CosPGD but only after 20 attack iterations. While CosPGD is significantly better for lower numbers of iterations. SegPGD nearly closing the gap in performance, despite taking 20 attack iterations, speaks to the credit of needing pixel-wise scaled adversarial attacks.
>
>
> For the task of image restoration, 20 attack iterations take a significantly long amount of time to evaluate.
> Thus we propose CosPGD, **an efficient attack** which as discussed in Section 4.3 can be very useful in predicting a new model’s robustness efficiently.
> Additionally, we would like to request the reviewer to reconsider the following snippet for Semantic segmentation from **Table 4 Section B.3 in the supplementary material**. Here we observe, that despite the numerical difference in performance at 20 iterations being low, the maximum difference being 3.19%, its significance is very high. CosPGD is bringing down the performance of multiple powerful segmentation models to almost 0% in just 20 attack iterations.
>
> |Model|Attack|mIoU at 20 attack iterations|mAcc at 20 attack iterations|
> |:----|:----|:----|:----|
> |UNet|SegPGD|2.98|14.24|
> | |CosPGD|0.06|0.38|
> |PSPNet|SegPGD|1.82|7.39|
> | |CosPGD|0.04|0.11|
> |DeepLabV3|SegPGD|3.31|12.4|
> | |CosPGD|0.12|0.48|
>
> Please note as well that the remaining prediction quality after SegPGD and CosPGD are both very low for 20 attack iterations, and CosPGD is still consistently better.
>
> 3. SegPGD can not be used on Optical Flow for the reasons detailed in Answer 1 unless the method is modified by defining a threshold. Similar to the results we provide for SegPGD for restoration, we can also report for optical flow as detailed below:
>
> Table 6 in Section C.1 in the appendix of the revised version (Table 4 in original submission). A Lower(↓) epe wrt to Target and a Higher( ↑) epe wrt to Initial signifies a stronger attack.
>
> |Attack Iterations|KITTI2015| | | | | |MPI Sintel| | | | | | | | | | | |
> |----:|----:|----:|----:|----:|----:|----:|----:|----:|----:|----:|----:|----:|----:|----:|----:|----:|----:|----:|
> | | | | | | | |Clean| | | | | |Final| | | | | |
> | |PGD| |CosPGD| |SegPGD| |PGD| |CosPGD| |SegPGD| |PGD| |CosPGD| |SegPGD| |
> | |Target↓|Initial ↑|Target↓|Initial ↑|Target↓|Initial ↑|Target↓|Initial ↑|Target↓|Initial ↑|Target↓|Initial ↑|Target↓|Initial ↑|Target↓|Initial ↑|Target↓|Initial ↑|
> |10|10.5|22.1|9|23.4|11.08|21.36|3.4|11.2|2.9|11.4|3.51|11.16|3.1|11.3|2.6|11.5|3.13|11.32|
> |20|8.1|24.6|6.5|25.8|7.76|24.55|2.8|11.7|2|12.1|2.97|11.61|2.5|11.8|1.6|12.1|2.62|11.7|
> |40|7.3|25|4.8|27.4|7.53|24.89|2.8|11.7|1.6|12.4|2.66|11.8|2.6|12.3|1.3|12.3|2.4|11.83|
>
> However, please note again that SegPGD was not conceived for optical flow attacks and the low performance is to be expected since SegPGD is not designed for regression tasks.
>
> 4. CosPGD is stronger when few iterations are considered and therefore provides reliable results at a lower compute budget. The purpose of adversarial attacks is to reveal model weaknesses with as few iterations as possible. Therefore, we understand that an attack that outperforms others when run for only a few attack iterations is particularly valuable. In particular, when considering adversarial training, high attack iterations are extremely expensive and 3 attack iterations would be a typical value.
>
> We hope we were able to answer all your questions to your satisfaction. Please let us know if you have further questions or concerns.
>
> Best Regards
>
> Authors of Paper #909

---

> ### Author Response · Authors · 2023-11-20
> **Gentle Reminder**
>
> Dear Reviewer E2Ss,
>
> We were curious if you happened to find the time to read our official comments and incorporated changes in the revised version. If so, we would be glad to answer any further questions or doubts you might have.
>
> Best Regards,
>
> Authors of Paper #909

---

> ### Author Response · Authors · 2023-11-22
> **Authors' reply to 'After Rebuttal'**
>
> Dear Reviewer E2Ss,
>
> Thank you for acknowledging the strength of our work in terms of experiments and analysis.
> We would appreciate it if you would consider the following,
>
> Both SegPGD and CosPGD propose to modify the update step of the original PGD applied to each pixel, as given in our equation (1).
>
> $sign\nabla_{\boldsymbol{X}}L$
>
>  Here, we write L to denote the original model loss with respect to the current model $f$'s prediction based on an adversarial sample
> $\boldsymbol X$
>
> and the one-hot encoded ground truth $Y$.
>
> For segmentation, i.e. in SegPGD $L$ is a cross-entropy loss as specified in their equation (2) . In our paper, we also use a cross entropy loss when considering segmentation, but $L$ can in principle be any loss of the respective original model, as in PGD. For optical flow, we use an optical flow loss, to be consistent with PGD.
>
> In SegPGD, the above update is modified to
>
> $sign\nabla_{\boldsymbol{X}}(\frac{1-\lambda}{N}\sum_{i\in P^T} L_i  + \frac{\lambda}{N}\sum_{k\in P^F} L_k)$
>
> where  $P^T$
> is the set of correctly classified pixels and  $P^F$
> is the set of wrongly classified pixels, $N$ is the total number of pixels, and $\lambda$ is a scaling factor between the two parts of the loss that is set heuristically. See their equation (4) for details.
>
> For positive $\lambda$, this equation could be rewritten as
>
> $sign\nabla_{\boldsymbol{X}}(\frac{1}{N}\sum_{i\in P^T\cup P^F} (1- |\lambda - |(argmax(f(\boldsymbol{X}))-Y|/2|) L_i) $
>
> for adversarial examples
> $\boldsymbol{X}$
>
> i.e. $|\lambda - |(argmax(f(\boldsymbol{X}))-Y|/2|$ equals $1-\lambda$ for incorrect predictions, it equals $\lambda$ for correct predictions.
>
> You can consider this representation of SegPGD to be the starting point of our argument: no matter what loss to use for L, we argue that the weighting of the pixel-wise loss with this weight after the argmax is an issue: it limits SegPGD to applications where the correctness of the prediction can be evaluated in a binary way, and it disregards the actual prediction scores. This is why CosPGD proposes in our equation (5) proposes to use a continuous measure of correctness instead:
>
> $sign\nabla_{\boldsymbol{X}}(\frac{1}{N}\sum_{i\in P^T\cup P^F} cos(softmax(f(\boldsymbol{X})), Y) L_i) $
>
> Please note that this facilitates CosPGD to operate on the segmentation scores instead of the final argmax predictions. Positive side aspects are that we do not need to set any heuristic parameter $\lambda$ and that we can directly apply this same procedure for a wide variety of tasks beyond segmentation. The experiments we show demonstrate the significant benefit of considering the actual prediction scores w.r.t. the cosine similarity of their softmax to the ground truth.
> We empirically show the benefit of CosPGD over SegPGD in a wide variety of experiments.
>
> Thus, we do not modify the loss, neither does SegPGD, the loss is the respective loss of the downstream task.
> **We are replacing the scaling in SegPGD with a closed-form continuous setting i.e. cosine similarity**.
>
> We agree that SegPGD, with some modifications, can be extended to other downstream tasks. And we proposed CosPGD, which goes beyond SegPGD and is significantly stronger than SegPGD on the task for which SegPGD was proposed.
> And, as shown in our experiments, **CosPGD extends to other tasks much better than the adaptations of SegPGD to those tasks**.
>
> Additionally, adversarial attacks are time and resource consuming. Thus, the availability of an efficient adversarial attack that requires merely 3 attack iterations to efficiently gauge a model’s relative robustness is indeed a significant contribution to the field. As discussed in the paper for each downstream task considered, CosPGD is able to expose model vulnerabilities that were previously unknown even with SegPGD.
>
> Moreover, CosPGD requires merely 3 attack iterations during adversarial training to train a significantly more robust model.
>
>
> Does this address your concern with respect to the difference of CosPGD to SegPGD and the novel contribution of this work?
>
> Best Regards
>
> Authors of Paper #909

---

### Official Review · Reviewer_kcAq · 2023-10-29

**Soundness:** 3 good
**Presentation:** 2 fair
**Contribution:** 3 good
**Rating:** 5
**Confidence:** 4

**Summary:**

This paper proposes CosPGD, a unified white-box adversarial attack aiming to any pixel-wise prediction task based on the cosine similarity between the distributions over the predictions and ground truth. The effectiveness of the method is demonstrated through a series of experiments across multiple tasks including semantic segmentation, optical flow and image denoising.

**Strengths:**

First and foremost, in comparison to the recently introduced SegPGD, CosPGD demonstrates a considerably more pronounced adversarial attack impact in semantic segmentation tasks. Notably, what sets CosPGD apart is its applicability beyond segmentation-specific tasks when compared to SegPGD. CosPGD serves as a versatile attack method applicable to any pixel-wise prediction task, boasting efficient deployment capabilities and superior efficacy in contrast to the general PGD method.

**Weaknesses:**

Section 4.3's content warrants appropriate adjustment. This section primarily showcases the superior degradation effect of CosPGD on NAFNet in comparison to PGD and SegPGD (particularly at low attack iterations). However, this evidence alone may not adequately support the assertion that "CosPGD can efficiently enhance a new model's robustness." To convincingly substantiate this claim, the authors should present more compelling evidence within the main body of the paper, rather than relegating it to the appendix. It is particularly essential to include results from the denoising task (as presented in Appendix D2).

**Questions:**

1. Although CosPGD exhibits substantial improvements over SegPGD in terms of attack efficacy and generality, it is worth noting that SegPGD also contributes significantly to enhancing model robustness through adversarial training. The absence of corresponding experiments makes it challenging to completely establish the effectiveness of this aspect.

2. An inquiry arises regarding the rationale behind the author's choice of an optical flow experiment to evaluate the versatility of CosPGD. The choice of optical flow as a benchmark should be substantiated by explaining how the characteristics of this task effectively highlight the advantages of CosPGD. Furthermore, additional experiments should be incorporated to showcase CosPGD's performance in various image restoration tasks, such as single image deraining, to bolster its claims further.

3. It seems like the authors need to reorganize the contribution of the paper, since the core of the paper is actually a general improvement on adversarial training for pixelwise classification tasks.

---

> ### Author Response · Authors · 2023-11-15
> **Authors' rebuttal to weakness and questions**
>
> Dear Reviewer kcAq,
>
> Thank you very much for your insightful review. We hope to answer your questions in a satisfactory manner.
>
> **Answer to weakness:**
>
> Please refer again to our section 4.3. We never make the claim that  "CosPGD can efficiently enhance a new model's robustness".
> Instead, we propose CosPGD as an effective adversarial attack that can evaluate a model’s relative robustness correctly even with low attack iterations (iterations<=3).
> Thus our claim is: “CosPGD can be very useful in **predicting** a new model’s robustness efficiently.” Would you agree that this claim is sufficiently substantiated by our experiments (including the new results we provide in the rebuttal), or is there anything else you would like to see in particular?
>
> **Answers to Questions:**
>
> 1. Adversarial attacks have a purpose beyond adversarial hardening: the evaluation of existing methods in terms of stability/robustness.  We therefore focused on evaluating CosPGD across diverse conditions. In the following, we also report a comparison in terms of adversarial training. Here the models were trained with the “Training Attack” with 3 attack iterations, $\alpha$=0.01 and $\epsilon\approx\frac{8}{255}$ with a 50%-50% minibatch split, meaning only 50% of the samples in a batch were adversarially perturbed. Then, the adversarially trained models were evaluated using “Testing Attack” with multiple attack iterations are shown in the following table (here we report only the mIoU(%) for better readability):
>
> |Model|Training Attack|Testing Attack|3 attack iterations|5 attack iterations|10 attack iterations|20 attack iterations|40 attack iterations|100 attack iterations|
> |:---:|:---:|:---:|:---:|:---:|:---:|:---:|:---:|:---:|
> |UNet|None|PGD|23.18|14.58|8.21|5.57|4.14|3.6|
> | |PGD|PGD|29.26|21.28|13.74|9.29|7.47|6.38|
> | |SegPGD|PGD|31.77|22.77|14.86|11.03|8.98|7.45|
> | |**CosPGD**|PGD|**47.35**|**43.75**|**38.1**|**34.33**|**32.28**|**30.55**|
> | |None|SegPGD|12.38|7.75|4.46|2.98|2.2|1.55|
> | |PGD|SegPGD|29.38|21.31|13.77|9.39|7.45|6.38|
> | |SegPGD|SegPGD|31.69|22.47|14.82|10.9|9.09|7.33|
> | |**CosPGD**|SegPGD|**47.16**|**43.85**|**37.64**|**33.99**|**31.91**|**30.48**|
> | |None|CosPGD|9.67|3.71|0.61|0.06|0.03|0.01|
> | |PGD|CosPGD|29.23|21.09|13.49|9.28|7.36|6.29|
> | |SegPGD|CosPGD|31.53|22.46|14.81|10.86|9.2|7.28|
> | |**CosPGD**|CosPGD|**47.07**|**43.95**|**37.64**|**34.01**|**32.0**|**30.55**|
>
>
> We include this table in the revised version of the paper as Table 5 in Section B.4
>
> Additionally, we include Figure 11 in Section B.4 in the revised version of the paper, in this figure we show the segmentation masks predicted by UNet after being adversarially trained. We observe that even after 100 attack iterations, the model adversarially trained using CosPGD is making reasonable predictions.
> However, the model trained with SegPGD is merely predicting a blob.
>
>
> 2. Could you please rephrase this question? We are not sure if we understand it correctly. Our rationale is that previous methods for attacks on pixel-wise prediction tasks always only focused on a single task, i.e. there has been SegPGD for semantic segmentation and PCFA for Optical Flow. CosPGD is more general than both and can therefore be applied to both. In Section 4.2 Optical Flow: we explain in detail how CosPGD is exposing model vulnerabilities that were unknown before.
> To further showcase the generality of the approach, we also evaluate it on image restoration for recent SotA methods. Does this address your question?
> Regarding deraining, we attempted to evaluate CosPGD on the SotA [a]. Yet, the model is too large to fit on our hardware when access to model gradients is needed (which is the case for white-box attacks). The model [a] was trained on 32 NVIDIA Tesla V100. If you can refer us to a more light-weight model of your choice, we will gladly provide the evaluation.
> [a] Chen et al., Pre-Trained Image Processing Transformer (IPT), 2021.
>
>
> 3. There seems to be a misconception: as specified in the contribution section, the contribution is not the improvement of adversarial training but proposing the first adversarial attack that provides a unified evaluation for diverse pixel-wise prediction tasks. Of course, adversarial attacks can be used for adversarial training (see our response to your Q1 for such results) and better adversarial attacks often result in better adversarial training. Our results above show that this is also the case for CosPGD.
>
> We hope we were able to answer all your questions to your satisfaction. Please let us know if you have further questions or concerns.
>
> Best Regards
>
> Authors of Paper #909

---

> ### Author Response · Authors · 2023-11-20
> **Gentle Reminder**
>
> Dear Reviewer kcAq,
>
> We were curious if you happened to find the time to read our official comments and incorporated changes in the revised version. If so, we would be glad to answer any further questions or doubts you might have.
>
> Best Regards,
>
> Authors of Paper #909

---

> > ### Author Response · Authors · 2023-11-22
> > **Last few hours of Authors-Reviewers discussion**
> >
> > Dear Reviewer kcAq,
> >
> > Following your suggestions, we have additionally included new results in the revised version of the paper.
> >
> > Since the authors' discussion phase is closing soon, we would like to ask you if you have any further questions or any questions unanswered.
> >
> > If not, we would like to request you re-evaluate your current score and raise it accordingly, as we believe we have answered all your concerns and incorporated your suggested improvements.
> >
> > Best Regards
> >
> > Authors of Paper #909

---

### Official Review · Reviewer_YMdT · 2023-11-01

**Soundness:** 3 good
**Presentation:** 4 excellent
**Contribution:** 3 good
**Rating:** 8
**Confidence:** 5

**Summary:**

This paper concentrates on adversarial attacks tailored for pixel-wise prediction tasks such as semantic segmentation, optical flow prediction, and image restoration.
It uncovers that PGD, a method commonly used in image classification, is not efficient for pixel-wise prediction tasks, and SegPGD, a method designed for semantic segmentation, is not applicable to other pixel-wise tasks.
The paper introduces CosPGD, an efficient white-box adversarial attack specifically designed for pixel-wise prediction tasks. It utilizes cosine similarity between prediction distributions and ground truth (or target, in the case of targeted attacks) to weight the loss value of each pixel, enabling more effective and nuanced attacks.
Experimental results across various datasets and settings demonstrate CosPGD's superiority and versatility in assessing the robustness of models for pixel-wise prediction tasks.

**Strengths:**

1. The proposed CosPGD is a relatively simple modification of SegPGD, yet it significantly enhances effectiveness across multiple datasets. While SegPGD differentiates between pixels that are predicted correctly and those predicted incorrectly during the generation of adversarial examples, assigning different pre-defined weights to the loss terms of correctly and incorrectly predicted pixels, CosPGD replaces these pre-defined weights with cosine similarities between the predictions and ground truth at each pixel. Experimental results demonstrate that this modification results in a more effective attack.

2. CosPGD is applicable to a variety of pixel-wise prediction tasks, including semantic segmentation, optical flow prediction, and image restoration. Unlike SegPGD, which is limited to pixel-wise classification tasks, CosPGD can be readily extended to both pixel-wise classification and regression tasks. Experimental results confirm the effectiveness of CosPGD on several pixel-wise prediction tasks.

3. There are abundant ablation experiments regarding hyper-parameters such as perturbation bounds, step sizes, and iteration steps, all of which verify the effectiveness of CosPGD compared to previous methods like PGD and SegPGD.

**Weaknesses:**

1. The paper does not provide sufficient comparisons and discussions related to recent works in pixel-wise prediction tasks, such as Qu et al. [1], and other applicable attacks in image classification, like C&W [2], and MI-FGSM [3].

[1] Qu et al. "A Certified Radius-Guided Attack Framework for Image Segmentation Models."
[2] Carlini et al. "Towards Evaluating the Robustness of Neural Networks."
[3] Dong et al. "Boosting Adversarial Attacks with Momentum."

2. Why does using cosine similarity as a weight (in CosPGD) outperform predefined weights (in SegPGD)? Is there a detailed explanation?

3. Why does the paper adopt different settings for the three tasks: non-targeted attacks for semantic segmentation and image restoration, and targeted attacks for optical flow prediction? What about the performance of targeted attacks for semantic segmentation and image restoration?

4. The experimental results presented in Figures 14 and 15 make it challenging to discern the numerical values. Presenting the data in a tabular form would be more beneficial.

5.There is a lack of a detailed definition for $L$ in equations (1), (5), and (6).

**Questions:**

See in weakness

---

> ### Author Response · Authors · 2023-11-15
> **Additional evaluations on C&W and MI-FGSM and answers to the questions**
>
> Dear Reviewer YMdT,
>
> Thank you very much for your insightful review. We hope to answer your questions in a satisfactory manner.
>
> 1. Thank you for the suggestion. We provide the following evaluation on semantic segmentation on Pascal VOC 2012 using UNet for [2] and [3]. For C&W [2], we have been evaluating the default parameters (c=1) here but have a strong suspicion that tuning c will improve results. We experimented with c={0.5, 1, 2, 8} and it did not make an impact on the performance. Would you have a particular suggestion for the choice of c?
>
> For MI-FGSM [3] 100 iterations, the evaluation did not finish within 24 hours, which was the time limit for our GPU usage, but we can report all other iterations.
> We observe that CosPGD significantly outperforms all other attacks, across attack iterations and $l_p$ norms. For $l_{\infty}$-norm attacks $\alpha$=0.01 and $\epsilon \approx \frac{8}{255}$. We add these results to Table 4 in Section B.3 in the revised version of the paper.
>
> For $l_2$-norm attacks in the table, where applicable $\alpha$=0.2 and $\epsilon \approx \frac{128}{255}$.
> We include this table in the revised version of the paper as Table 3 in Section B.2.1
>
>  (here we report only the mIoU(%) for better readability)
>
> |Model|Attack|Norm|3 iterations|5 iterations|10 iterations|20 iterations|40 iterations|100 iterations|
> |:---:|:---:|:---:|:---:|:---:|:---:|:---:|:---:|:---:|
> |DeepLabV3|MI-FGSM|$l_{\infty}$|10.86|7.75|6.95|6.67|6.57|—|
> | |PGD|$l_{\infty}$|10.69|8|7.02|6.84|6.79|7.01|
> | |SegPGD|$l_{\infty}$|6.76|4.86|3.84|3.31|2.69|2.15|
> | |**CosPGD**|$l_{\infty}$|**4.44**|**1.84**|**0.69**|**0.12**|**0.08**|**0.005**|
> | |CW (c=1)|$l_2$|72.35|72.02|71.87|71.81|71.78|71.77|
> | |PGD|$l_2$|41.81|34.5|27.61|23.73|21.47|19.84|
> | |SegPGD|$l_2$|37.51|29.9|22.72|19.2|16.8|14.77|
> | |**CosPGD**|$l_2$|**36.17**|**27.12**|**18.68**|**14.35**|**12.23**|**10.97**|
>
> Also, thank you for the pointer to reference [1]. Our experiments focus on the wide applicability of CosPGD rather than on outperforming a particular, dedicated approach on a particular task. We will try to provide results comparing our method to [1] for the final version, which is however not trivial since [1] proposes a certified radius-guided attack framework and CosPGD is an epsilon-bounded attack. In any case, we cite the paper in the revised version. Please note that according to the ICLR guidelines, https://iclr.cc/Conferences/2024/ReviewerGuide, publications from recent conferences (“published [...] within the last four months”: “on or after May 28, 2023”) are assumed to be contemporaneous work and a comparison is not required - this is the case for [1].
>
> Additionally, we came across another interesting related work that we have now cited in the related work of the revised version of the paper.
> Jia, J., Qu, W. and Gong, N., 2022. MultiGuard: Provably Robust Multi-label Classification against Adversarial Examples. Advances in Neural Information Processing Systems, 35, pp.10150-10163.
>
> 2. The most important conceptual difference between SegPGD and CosPGD is that the weighting with fixed weights is applied after the argmax operation in SegPGD. This removes important information and makes the attack optimization unstable. As a remedy, SegPGD uses a linear combination of the weights and non-weighted loss terms, where the combination weight is a heuristic. The use of the cosine similarity is therefore more informative: we compute the softmax of the class scores for segmentation to preserve the continuous prediction information. For the resulting positive valued prediction vectors, the cosine similarity is particularly suitable. Additionally, it extends well to pixel-wise regression tasks like optical flow estimation, image restoration etc. If the reviewer suggests, we can add this discussion to the final paper.

---

> > ### Author Response · Authors · 2023-11-15
> > **Further answers**
> >
> > 3. In the main paper, we show a selection of combinations to demonstrate the method’s versatility. There is the following reasoning for the selection we made:
> >
> >      - Previous works (SegPGD) only show their performance in the untargeted setting. Many previous works on adversarial attacks consider this to be the default.
> >
> >      - Since the default is an untargeted attack, we focus on untargeted attacks for restoration as well. The untargeted setting makes particular sense for semantic segmentation. There, one can assign for every pixel whether it is predicted correctly or incorrectly. Therefore, a reasonable aim for an attack is to turn as many pixel predictions incorrect as possible. For restoration, this also works because the attacked image as well as the restored image have bounded values.
> >
> >    - For optical flow, the previous SotA (PCFA) focused on targeted attacks - therefore, we consider targeted attacks for optical flow in the main paper. This makes sense because there is no binary decision for a prediction to be incorrect. In an untargeted setting for optical flow, the overall aim is achieved if the prediction is very wrong in a few locations (because the flow magnitude at every location is unbounded) or significantly wrong in many locations, making the evaluation ambiguous. Therefore, for optical flow, the aim of targeted training is better defined as turning the prediction to be as close as possible to the target in all locations.
> >
> > For completeness, we have however also reported results for untargeted attacks on optical flow in our original submission in section C2 (supplementary material), in Figures 12 and 13 in the revised version (Figures 11 and 12 in the original submission).
> >
> > Does this answer the question? For completeness, we will add evaluations on targeted attacks for segmentation in the final submission
> >
> > 4. Thank you for the suggestion. These values are indeed very similar to each other in many cases, and the plots are supposed to provide this information. When considering Figure 14 in the revised version (Figure 13 in the original submission, when considering targeted attacks as a whole), it is visible from the reported standard deviation that the difference is not always significant in this case. Yet, so far, we only have one run for the results in Figure 15 (target flow being 0 vector) and Figure 16(target flow being negative initial flow) in the revised version(Figures 14 and 15 in the original submission) and we do not observe any significant difference in numbers. We are currently running these experiments with multiple runs to be able to provide tabular results with standard deviations over multiple seeds for a better understanding.
> >
> > 5. Indeed, we did not specify a particular loss $L$ in equations (1), (5) and (6). In all cases, it is supposed to be the loss of the model at hand, which will be a different loss term for semantic segmentation than for optical flow and again a different one for restoration. Since we propose CosPGD as an attack that can be applied across diverse pixel-wise prediction tasks, $L$ can take different forms. We now specify in the revised submission that $L$ is assumed to be a one-differentiable function of the model prediction and the target, that defines the loss the model aims to minimize.
> >
> > We hope we were able to answer all your questions to your satisfaction.
> > Please let us know if you have further questions or concerns.
> >
> > Best Regards
> >
> > Authors of Paper #909

---

> > > ### Comment · Reviewer_YMdT · 2023-11-22
> > > **Feedback for Rebuttal**
> > >
> > > Thanks for your solid rebuttal. I will slightly raise the score

---

> > > > ### Author Response · Authors · 2023-11-22
> > > > **Authors' Gratitude**
> > > >
> > > > Dear Reviewer YMdT,
> > > >
> > > > Thank you very much for your feedback and for raising the score.
> > > > Please do let us know if you have any further questions or concerns, we would be glad to address them.
> > > >
> > > > Best Regards
> > > >
> > > > Authors of Paper #909

---

> ### Author Response · Authors · 2023-11-20
> **Additional Results**
>
> Dear Reviewer YMdT,
>
> Following your suggestion in Weakness #4, we have now included **Table 8 in the revised version** to be included in Section C.3 later, which is an extension to results from Figures 14, 15, and 16 in the revised version (were Figures 13, 14, and 15 in the original submission) over 3 random seeds in a tabular form.
>
> For ease of reading, in the following tables, we present Table 8: a comparison of the performance of CosPGD to PCFA as a targeted $l_2$-norm constrained attack for optical flow estimation over KITTI2015 and Sintel validation datasets using different optical flow models over 3 random seeds. Average $epe$ (AEE) values are compared, with respect to both, the **Target**  where a lower $epe$ indicates a better attack and **Initial flow prediction** (optical flow estimated by the model before any adversarial attack) where a higher $epe$ indicates a better attack.
> We compare over both targets used by [4], i.e. zero vector $\overrightarrow{0}$ and Negative of the Initial Flow.
> As observed earlier in the respective figures, **the performance of CosPGD is at-par with the recently proposed Optical Flow specific attack**.
>
>
> |Model| | || | KITTI 2015 | | | |
> |:---:|:---:|:---:|:---:|:---:|:---:|:---:|:---:|:---:|
> | | | Negative Flow Target | | ||Zero Vector Target|  | |
> | |AEE wrt initial| |AEE wrt Target| |AEE wrt initial| |AEE wrt Target| |
> | |CosPGD|PCFA|CosPGD|PCFA|CosPGD|PCFA|CosPGD|PCFA|
> |GMA|47.00 ± 0.40|47.08 ± 0.69|19.22 ± 0.53|19.20 ± 0.57|28.69 ± 0.12|28.67 ± 0.17|3.89 ± 0.09|3.89 ± 0.15|
> |PWCNet|33.13 ± 0.25|33.13 ± 0.26|12.01 ± 0.20|12.02 ± 0.22|19.13 ± 0.04|18.96 ± 0.08|3.25 ± 0.08|3.47 ± 0.14|
> |RAFT|48.83 ± 0.35|48.93 ± 0.29|17.97 ± 0.29|17.81 ± 0.27|29.09 ± 0.03|29.17 ± 0.11|3.75 ± 0.05|3.63 ± 0.10|
> |SpyNet|12.10 ± 0.02|12.08 ± 0.05|16.47 ± 0.03|16.44 ± 0.05|9.00 ± 0.01|9.01 ± 0.03|5.31 ± 0.01|5.35 ± 0.06|
>
>
> |Model| | | | Sintel (clean)| | | | |
> |:---:|:---:|:---:|:---:|:---:|:---:|:---:|:---:|:---:|
> | | | Negative Flow Target|  | | | Zero Vector Target| | |
> | |AEE wrt initial| |AEE wrt Target| |AEE wrt initial| |AEE wrt Target| |
> | |CosPGD|PCFA|CosPGD|PCFA|CosPGD|PCFA|CosPGD|PCFA|
> |GMA|29.25 ± 0.38|29.05 ± 0.38|8.58 ± 0.34|8.82 ± 0.37|16.87 ± 0.14|16.76 ± 0.11|1.75 ± 0.15|1.85 ± 0.10|
> |PWCNet|20.57 ± 0.21|20.43 ± 0.21|13.20 ± 0.13|13.21 ± 0.29|12.20 ± 0.21|12.18 ± 0.07|4.87 ± 0.17|4.75 ± 0.12|
> |RAFT|29.01 ± 0.11|29.20 ± 0.01|7.67 ± 0.11|7.47 ± 0.05|16.42 ± 0.03|16.46 ± 0.05|1.69 ± 0.04|1.65 ± 0.06|
> |SpyNet|13.08 ± 0.01|13.17 ± 0.03|18.75 ± 0.02|18.76 ± 0.06|9.69 ± 0.01|9.75 ± 0.07|6.40 ± 0.05|6.35 ± 0.00|
>
>
> |Model| | | | Sintel (final)| | | | |
> |:---:|:---:|:---:|:---:|:---:|:---:|:---:|:---:|:---:|
> | | | Negative Flow Target| | | | Zero Vector Target| ||
> | |AEE wrt initial| |AEE wrt Target| |AEE wrt initial| |AEE wrt Target| |
> | |CosPGD|PCFA|CosPGD|PCFA|CosPGD|PCFA|CosPGD|PCFA|
> |GMA|32.11 ± 0.20|32.04 ± 0.24|4.57 ± 0.22|4.64 ± 0.24|17.34 ± 0.07|17.31 ± 0.11|0.53 ± 0.07|0.54 ± 0.11|
> |PWCNet|23.00 ± 0.30|23.01 ± 0.06|10.84 ± 0.28|10.75 ± 0.05|13.61 ± 0.10|13.44 ± 0.14|3.52 ± 0.13|3.66 ± 0.12|
> |RAFT|32.72 ± 0.22|32.72 ± 0.14|3.71 ± 0.21|3.75 ± 0.13|17.38 ± 0.04|17.36 ± 0.03|0.55 ± 0.09|0.50 ± 0.03|
> |SpyNet|16.51 ± 0.01|16.55 ± 0.06|16.52 ± 0.01|16.47 ± 0.05|11.56 ± 0.01|11.59 ± 0.03|4.97 ± 0.01|4.97 ± 0.01|
>
> Please do let us know if you have any further questions or concerns, we would be glad to address them.
>
> Best Regards
>
> Authors of Paper #909
>
> [4] Schmalfuss, Jenny, Philipp Scholze, and Andrés Bruhn. "A perturbation-constrained adversarial attack for evaluating the robustness of optical flow." European Conference on Computer Vision. Cham: Springer Nature Switzerland, 2022.

---

### Author Response · Authors · 2023-11-15
**General Reply: Differentiating CosPGD from SegPGD**

Dear AC and Reviewers,

We thank all reviewers and the area chair for reviewing our paper and providing knowledgeable feedback. In particular, we appreciate that all reviewers acknowledge the benefit of our proposed approach in terms of attack effectiveness, especially its benefits over SegPGD for semantic segmentation.
In the following, we summarise again how our proposed approach CosPGD can be delineated from SegPGD which, when looking at the core equations, is closely related.

There are several important differences between CosPGD and SegPGD. Most importantly, there is a conceptual difference: SegPGD aims to scale the loss with the pixel-wise correctness of the model prediction after applying the **argmax** operation. **CosPGD aims to scale the loss with the similarity between posterior and target distributions.**
SegPGD has several limitations that are overcome by CosPGD:

1. SegPGD requires the use of a linear combination of the full model loss and the weighted loss to allow for stable training. The weights of the linear combination are assigned in a heuristic way.

2. SegPGD can only be applied if it can be determined in a binary way, for every pixel, if the prediction is correct. For optical flow or image restoration, the prediction values are not categorical values  - the prediction of a single value is neither correct nor wrong because it approximates a continuous function.
Therefore, SegPGD does not naturally extend to these tasks. (To compare to SegPGD on optical flow or image restoration, one could define a threshold from when a prediction is “good enough” but this has not been explored in their paper. We evaluate the setting anyway on image restoration.)

3. Since SegPDG operates on the argmax, it does not fully leverage the predictions of the model.

In contrast, CosPGD leverages the model predictions **before** taking the argmax. By taking the softmax of the semantic segmentation predictions, the class scores are transformed such that the cosine similarity becomes a suitable similarity measure. Therefore, in contrast to SegPGD, our method does not need any heuristic to remix losses and directly outperforms SegPGD.
Furthermore, **in contrast to SegPGD, CosPGD is not limited to the evaluation of segmentation models**. We show that it can be applied to other tasks as well, e.g. optical flow, and image restoration. SegPGD can not directly be used there.
Thus, although the main equations look similar, the conceptual difference is significant and leads to strong advantages.

We have included suggested changes from the reviewers and we highlight the changes in the revision submission with blue text colour.
We address all further questions by the reviewers one by one. We look forward to a healthy discussion.

Best Regards

Authors of Paper #909

---

> ### Author Response · Authors · 2023-11-22
> **Further differences highlighted using mathematical formulation**
>
> Dear Reviewers and AC,
>
> Thank you for acknowledging the strength of our work in terms of experiments and analysis.
> Since there is an ongoing discussion on alternative losses and the difference of CosPGD to SegPGD, we elaborate on this question from a **mathematical formulation** in the following, which, as we hope, will clarify the difference quite clearly.
>
> Both SegPGD and CosPGD propose to modify the update step of the original PGD applied to each pixel, as given in our equation (1).
>
> $sign\nabla_{\boldsymbol{X}}L$
>
> Here, we write L to denote the original model loss with respect to the current model $f$'s prediction based on an adversarial sample
> $\boldsymbol X$
>
> and the one-hot encoded ground truth $Y$.
>
> For segmentation, i.e. in SegPGD $L$ is a cross-entropy loss as specified in their equation (2) . In our paper, we also use a cross entropy loss when considering segmentation, but $L$ can in principle be any loss of the respective original model, as in PGD. For optical flow, we use an optical flow loss, to be consistent with PGD.
>
> In SegPGD, the above update is modified to
>
> $sign\nabla_{\boldsymbol{X}}(\frac{1-\lambda}{N}\sum_{i\in P^T} L_i  + \frac{\lambda}{N}\sum_{k\in P^F} L_k)$
>
> where  $P^T$
> is the set of correctly classified pixels and  $P^F$
> is the set of wrongly classified pixels, $N$ is the total number of pixels and $\lambda$ is a scaling factor between the two parts of the loss that is set heuristically. See their equation (4) for details.
>
> For positive $\lambda$, this equation could be rewritten as
>
> $sign\nabla_{\boldsymbol{X}}(\frac{1}{N}\sum_{i\in P^T\cup P^F} (1- |\lambda - |(argmax(f(\boldsymbol{X}))-Y|/2|) L_i) $
>
> for adversarial examples
> $\boldsymbol{X}$
>
> i.e. $|\lambda - |(argmax(f(\boldsymbol{X}))-Y|/2|$ equals $1-\lambda$ for incorrect predictions, it equals $\lambda$ for correct predictions.
>
> You can consider this representation of SegPGD to be the starting point of our argument: no matter what loss to use for L, we argue that the weighting of the pixel-wise loss with this weight after the argmax is an issue: it limits SegPGD to applications where the correctness of the prediction can be evaluated in a binary way, and it disregards the actual prediction scores. This is why CosPGD proposes in our equation (5) proposes to use a continuous measure of correctness instead:
>
> $sign\nabla_{\boldsymbol{X}}(\frac{1}{N}\sum_{i\in P^T\cup P^F} cos(softmax(f(\boldsymbol{X})), Y) L_i) $
>
> **Thus we are replacing the scaling in SegPGD with a closed form continuous setting i.e. cosine similarity.** Please note that this facilitates CosPGD to operate on the segmentation scores instead of the final argmax predictions. Positive side aspects are that we do not need to set any heuristic parameter $\lambda$ and that we can directly apply this same procedure for a wide variety of tasks beyond segmentation. The experiments we show demonstrate the significant benefit of considering the actual prediction scores w.r.t. the cosine similarity of their softmax to the ground truth.
> We empirically show the benefit of CosPGD over SegPGD in a wide variety of experiments.
>
> Moreover, CosPGD requires merely 3 attack iterations during adversarial training to train a significantly more robust model. We include these results in Table 5 in Section B.4.
>
> We hope this clears the remaining doubts regarding the novelty of CosPGD wrt to SegPGD.
>
> Best Regards
>
> Authors of Paper #909

---

### Author Response · Authors · 2023-11-23
**Post Discussion Summary**

Dear Reviewers and AC,

We sincerely thank all reviewers for their constructive feedback and comments. We are pleased to see that the reviewers found our method “significantly enhances effectiveness across multiple datasets” (Reviewer YMdT) and “serves as a versatile attack method applicable to any pixel-wise prediction task, boasting efficient deployment capabilities and superior efficacy in contrast to the general PGD method” (Reviewer kcAq). We appreciate that all reviewers acknowledge the benefit of our proposed approach in terms of attack effectiveness, especially its benefits over SegPGD for semantic segmentation.   In particular, we appreciate the strong support for our paper by Reviewer YMdT, after the rebuttal.

We are also glad that after the discussion period, not only Reviewer-YMdT increased the score from 6 to 8, underlining the benefit of our work, but also Reviewer-E2Ss increased the score from 3 to 5, acknowledging our additional experimental results. The remaining concern that, as we understand from the discussion, keeps Reviewer-EsSs, Reviewer-kcAq, and Reviewer-3VUb  from further increasing their scores is the theoretical delineation of our work from SegPGD, while all acknowledge the practical benefit. To address this point, we have described the mathematical difference of our work to SegPGD in our recent post from Nov. 22nd titled “Further differences highlighted using the mathematical formulation”.

We hope that this can resolve the remaining concern and can be taken into account in the final discussion.

We genuinely thank all the reviewers for their insightful suggestions, which have significantly improved the quality of our manuscript during the revision process.



Best Regards

Authors of Paper #909

---

### Meta-Review · Area_Chair_WmiF · 2023-12-09

**Metareview:**

The meta-reviewer has carefully read the paper, reviews, rebuttals, and discussions between authors and reviewers. The meta-reviewer agrees with the reviewers that this paper is at the acceptance boundary of ICLR.  Although YMdT increased the score from 6 to 8, and E2Ss increased the score from 3 to 5, they still have concerns, such as (1) While the rebuttal claims that "there is no other attack method scaling the loss pixel-wise using similarity between the posterior and target distributions", which seems a bit trivial and cannot be regarded as a main contribution to this field, and (2) Although SegPGD can not be directly applied to image restoration, it can be adapted to other supervised learning tasks by doing some simple modifications. E2Ss increased the score because the authors provided lots of experiments in the rebuttal. The meta-reviewer believes that putting these lots of new experiments and analysis in the paper is not an easy task and may change the current form of the submission a lot. The meta-reviewer agrees with the reviewers this paper can be a good submission in the next venue after incorporating the new experiments and analysis and re-organize the presentation of the paper.

**Justification For Why Not Higher Score:**

N/A

**Justification For Why Not Lower Score:**

N/A

---

### Decision · Program_Chairs · 2024-01-16

Reject